# Beekeeping Genetic Resources and Retrieval of Honey Bee *Apis mellifera* L. Stock in the Russian Federation: A Review

**DOI:** 10.3390/insects12080684

**Published:** 2021-07-29

**Authors:** Olga Frunze, Anna Brandorf, Eun-Jin Kang, Yong-Soo Choi

**Affiliations:** 1Department of Agricultural Biology, National Institute of Agricultural Science, Wanju 55365, Korea; frunzeo1@korea.kr (O.F.); kangeunjin1@korea.kr (E.-J.K.); 2Federal State Budgetary Scientific Institution “Federal Beekeeping Research Center”, Ministry of Science and Higher Education of Russia, 391110 Rybnoye, Russia; apis_mellifera_mellifera_l@mail.ru

**Keywords:** Russian beekeeping, honey bee losses, bee breeding, *Apis mellifera* breeds, bee mite resistance

## Abstract

**Simple Summary:**

The loss of honey bees poses a significant problem for the beekeeping industry. Opportunities to recover the stock of honey bees are crucial in areas with drastic losses. In this article, we describe known native and bred *Apis mellifera* L. stocks of honey bees in the Russian Federation and their characteristics, identified using morphometric and genetic methods. We review the experience of Russian breeders and other breeders in breeding initial *A. mellifera* Far East honey bees with inherited traits for *Varroa* resistance. We also describe the bred types of honey bee *A. m. mellifera* L., *A. m. caucasia* Gorb., *A. m. carpatica* Avet. that offer potential in the aim to recover honey bee populations after losses. Our findings indicate that it is necessary to avoid honey bee losses by breeding resistant honey bees. Several bred types of honey bees from the dataset showed high performance in terms of overwintering, resistance to *Varroa destructor*, *Acarapis woodi*, and *Nosema* infections, and little or no swarm conditions. This review shows the potential to increase selection efforts in the breeding of resistant *Apis mellifera* L. honey bee populations in the Russian Federation and throughout the world.

**Abstract:**

The loss of honey bees has drawn a large amount of attention in various countries. Therefore, the development of efficient methods for recovering honey bee populations has been a priority for beekeepers. Here we present an extended literature review and report on personal communications relating to the characterization of the local and bred stock of honey bees in the Russian Federation. New types have been bred from local colonies (*A. mellifera* L., *A. m. carpatica* Avet., *A. m. caucasia* Gorb.). The main selection traits consist of a strong ability for overwintering, disease resistance and different aptitudes for nectar collection in low and high blooming seasons. These honey bees were certified by several methods: behavioral, morphometric and genetic analysis. We illustrate the practical experience of scientists, beekeepers and breeders in breeding *A. mellifera* Far East honey bees with *Varroa* and tracheal mite resistance, which were the initial reasons for breeding the *A. mellifera* Far Eastern breed by Russian breeders, Russian honey bee in America, the hybrid honey bee in Canada by American breeders, and in China by Chinese beekeepers. The recent achievements of Russian beekeepers may lead to the recovery of beekeeping areas suffering from crossbreeding and losses of honey bee colonies.

## 1. Introduction

Beekeeping has an ancient and varied history, with evidence that honey bees have been kept for thousands of years and that honey bee husbandry practices have not changed in millennia [1]. Honey bees have been used for the production of honey, beeswax, royal jelly, propolis and pollen. Honey bee pollination is considered essential for the production of seeds and vegetables [2]. Bees contribute to one-third of the products consumed by humans, and equally pollinate 84% of the plants necessary for life [3]. However, in 1906 the loss of honey bees was reported in the small Isle of Wight off the south coast of England. Honey bee colonies died out, with honey bees seen crawling from the hive unable to fly [4]. The loss of honey bee colonies has continued and has been reported by scientists since the late 20th century, with the USA having about 30% of losses, 1.8–53% in Europe, 10–85% in the Middle East, 25% in Japan, and 20% in Russia [4,5,6,7,8,9,10]. This disorder is characterized by the loss of adult honey bees from hives, leaving behind the brood (larvae and pupae). Since the remaining worker bees cannot care for the brood, this leads to colony collapse. Usually, the dead honey bees are not found near their colonies [11]. Researchers from 11 countries within the COLOSS research network concluded that the main reasons for honey bee colony loss were *Varroa destructor* (38%), difficulties with the queen (loss, drone layer, 17%), and *Nosema* (8%). Furthermore, the explanation for the decline of honey bees seems to be increasingly attributable, not listed above in COLOSS research, to *Phorid* parasitoids. Diptera *Phoridae* can cause the weakening, reduction, or the disappearance of honey bees, with damage caused by *Apocephalus borealis* in the United States [12] and by *Megaselia* ssp. in Europe [13], Africa [14,15], and Asia [16]. The first warning was reported in the Russian Federation in 2014 when the parasite *Megaselia scalaris* (Loew) was introduced with plants [17]. However, *Megaselia* ssp. (not *Megaselia scalaris*) were present in the native environment in the Russian Federation, but their life cycle was restricted by low temperatures [18]. The mass loss of honey bees caused by infestations of *Megaselia scalaris* has not yet been detected in Russian Federation [19,20,21]. All other reasons (starvation, robbing, unspecified winter loss, other diseases, unknown reasons) account for 37% of the losses [22]. Similar issues exist in beekeeping in the Russia Federation and other countries [23,24,25,26,27,28,29,30]. In addition, climate change threatens honey bees in terms of the lack or reduction of resources (nectar, pollen and honeydew) and the spread of *Varroa* and other pests [31,32]. Beekeeping management, chemical toxicosis and herbicide poisoning have also been highlighted as reasons for honey bee losses [19,33].

The reasons for honey bee losses and methods of assessment and protection were discussed widely at the International Scientific and Practical Conference on “Modern Problems of Beekeeping and Apitherapy” (Russia, Rybnoye, 2020), and the International conference on “Modern aspects of apitherapy” (Ukraine, Kiev, 2020). The environmental situation has deteriorated around the world during recent years. It was noted that the crossbreeding of honey bees, chemical treatments of plants, insects and factory emissions have played a major role in this deterioration. The accumulation of metal ions in honey bee bodies has influenced the health of honey bee populations in the Ryazan region [34]. E.K. Eskov, M.D. Eskova (2021) [35] analyzed honey bee losses in polluted areas as a result of the accumulation of pollutants in the honey bee body. Moreover, the performance of honey bees was assessed in terms of hygienic behavior, which was linked with morphometry characteristics (cubital index, tarsal index, length of proboscis, and the width of tergite 3 and wing veins), honey productivity, and *Varroa* infestations [36]. In addition, the effectiveness of biotechnological methods has been reported for the prevention of *Varroa* mites in honey bee populations, including heating and the use of mite traps with the naturally attractive properties of medicinal plants [37].

Currently, beekeepers breed local, foreign, or inbred honey bee colonies. Traditional beekeeping was based on the breeding of local honey bee resources in a native environment [38,39,40,41]. These honey bees were found worldwide, from the tropics to subarctic temperate zones, and on all continents except Antarctica. Researchers described twenty-seven honey bee subspecies of *A. mellifera* L. and nine honey bee subspecies of *A. cerana* F. [42]. In recent decades, beekeepers have been attempting to find “the best bee” for honey production [19]. As a consequence, foreign bees were introduced by beekeepers into non-native environments [20,43,44,45,46,47]. The traits of longevity, gentleness and high honey production were examined in local and foreign honey bees [44,48]. Sometimes, benefits were introduced into the local honey bee populations [19,43]. However, new diseases were also introduced along with foreign honey bees, which has been the cause of local and foreign honey bee colony losses.

With the exception of pure local colonies, bee breeders breed new (inbred) types from foreign or local honey bees using selection methods and comparing their traits with initial or domestic colonies [49]. Inbred *A. mellifera* Far Eastern breed (Russian-bred) honey bees were bred from *A. mellifera* Far East honey bees [50]. Furthermore, the latter subspecies of honey bees was also the initial colony used for the breeding of Russian honey bees (American-bred) in the USA, hybrids of Russian and Ontario honey bees in Canada [51], and *Varroa*-resistant *A. mellifera* Far Eastern honey bees in China (Chinese-bred) [52]. As a consequence, inbred honey bees in the USA, Canada, and China all originated from introduced *A. mellifera* Far East honey bees. Inbred honey bees have special traits in comparison with domestic colonies. De Gusman et al. (2001) [53] evaluated the effectiveness of *A. woodi* control treatments in the breeding of Russian honey bees (in inbred honey bees from *A. mellifera* Far East honey bee). They found that the strong resistance of inbred Russian honey bees (American-bred bees) to tracheal mites was in itself sufficient to minimize tracheal mite damage. Thus, the resistance to *Varroa* and tracheal mites displayed by the initial honey bees (*A. mellifera* Far East) were transmitted to Russian honey bees (American-bred) during breeding [54]. Thus, some advantages are displayed by local and inbred honey bees in comparison with foreign colonies.

This review focuses specifically on known local and inbred types of genetic resources found in honey bees in the Russian Federation which have been identified through the use of various methods (behavior, genetics, morphometry). A second focus of this paper introduces the results of breeding the same initial *A. mellifera* Far East honey bees in different environments: in the Russian Federation, in the USA, in Canada and in China.

## 2. Beekeeping Practices in the Russian Federation

Russian farmers have been engaging in beekeeping not only as their main activity but as a sideline activity since the 10th and 11th centuries [41,55]. It is presumed that beekeepers have developed indigenous knowledge through their accumulated experiences, which represents a valuable resource for the development of beekeeping. However, beekeeping in the Russian Federation depends on climate, since more than half of its territory is located in the northern zone. The country has eight federal districts: 1, North Western (9.87% of all Russian Federation); 2, Central (3.80%); 3, South (2.62%); 4, North Caucasian (1.51%); 5, Volga (6.07%); 6, Ural (10.62%); 7, Siberian (25.50%) and 8, Far East (42.00%) districts (Figure 1).

The largest part of the Russian Federation is the Far East, where *A. mellifera* L. beekeeping is less than 150 years old [57,58]. The local forest honey bee was described by Kozhevnicov in 1926 as *A. cerana* (in modern taxonomy). At that time, beekeepers did not breed *A. cerana* honey bees in apiaries [59,60]. Both honey bees survived at average temperatures below −15 °C in January (the coldest month), and 20 °C in June and August (the warmest month) [20]. Average annual temperature varies from −10 °C in the north to 6 °C in the south. The overwintering of honey bees lasts from October to mid-May (6 to 7.5 months). Excellent bee forage ensures significant honey yields by honey bees (*A. mellifera* and *A. cerana*). In the Far East, the excellent honey yield is similar in blooming periods compared to regions with linden trees [57].

In Siberia, beekeeping was introduced about 230 years ago, and overwintering of honey bees lasts from five to sixmonths (from November to April) [61]. The average temperature in January (the coldest month) is −19.2 °C, and in July (the warmest month) it is 18.1 °C. The frost-free period is 100–105 days for the vegetation of honey plants.

Another beekeeping zone in Russian Federation is the Ural district. Beekeeping in the south and central Ural zones developed 400 years ago. The honey bees survive at the average temperature in January (the coldest month), which is –18.0 °C, and 18.0 °C in July (the warmest month). The frost-free period is 110–120 days for the vegetation of honey plants [62,63]. The overwintering of honey bees lasts seven months (from October to April) [56]. Honey bees are not present in the north Ural zone, because the weather makes it very hard for them to survive.

The mild climates in the southern part of Russia promote honey bee development and are not described in this review. These climates formed the environments for populations of *A. mellifera* L. which were detected to possess specific traits and have been used by breeders to breed new types of honey bees in the Russian Federation.

### 2.1. Purposes of Beekeeping: Breeding and Production

Modern beekeeping in the Russian Federation has both breeding and productive aims. The honey bee management practices differ according to these two aims. The purpose of breeding in beekeeping is the storage of the genetic lines of honey bees and the breeding of new types. Productive beekeepers have the purpose of attaining higher harvests of honey, royal jelly and other products. High productivity has been reached by breeding hybrid honey bee colonies with the ability to survive long overwintering periods. Hybrid honey bee colonies are the result of mating one purebred queen and other purebred drones. Usually, beekeepers buy new hybrid honey bees after sustaining losses. Hence, breeding (using purebred methods) and productive (breeding hybrid honey bee colonies) beekeeping practices are connected through the breeding of purebred and hybrid honey bees of different lines. However, some beekeepers buy foreign subspecies of honey bees, but breeding beekeeping practice is restricted to apiary location. Thus, the random movement of colonies containing foreign subspecies of honey bees makes it impossible to use natural mating for purebred honey bee colonies within 20 km of these subspecies [16]. This is why artificial insemination was developed in breeding apiaries. In addition, the occasional problem of finding locations for breeding apiaries was solved by the isolation of breeding apiaries in the mountains and in national parks, far from personal and productive apiaries. The results of the random natural crossbreeding of honey bees led to the loss of purebred honey bee colonies, a decreased ability to survive long overwintering periods and low resistance to diseases. Preventive actions against the uncontrolled crossbreeding of honey bees in breeding apiaries have not yet been developed at the state level.

### 2.2. Conservation of A. mellifera L. Honey Bees

Scientists, in collaboration with employees of national parks, aim to provide the best conditions for the conservation of *A. mellifera* L bees. Examples of these locations include the Tongariro National Park in New Zealand [64], the Wyperfeld National Park in Australia [5], the Nyungwe National Park in Rwanda [65] and other areas with special protection [66,67]. In Russia, some of the national parks listed below have been used for the conservation of *A. m. mellifera* breeds. These include: Shulgan-Tash Park in Bashkortostan, Vishersky and Malinovii Hutor in the Perm region, Orlovsky Polesie in the Orlov region, and a reserve in Tatarstan. The honey bees of *A. m. caucasia* were bred in Krasnodarskiy province, whereas those of *A. m. carpatica* were bred in the Republic of Adygea (Figure 2).

The existence of national parks in Russia is regulated by state documents [6]. One of the rules stipulates that a reserve may be organized on a land area having a radius of 25 km with an apiary in the central part of a national park in which the total amount of purebred honey bees is not less than 200 colonies. Basically, these apiaries have used natural mating or artificial insemination of the queen as breeding techniques. More than 20 breeding apiaries have been certified in the Russian Federation, and these have distributed purebred bees to other apiaries for breeding [11,68].

### 2.3. Evaluation of Honey Bees

To restore the original gene pool of the honey bee, precise identification of the subspecies is required [62]. Researchers have developed methods such as assessments of physiology and biochemistry [69,70,71,72,73,74], behavior [75,76], productivity [77,78,79,80], morphometry [81,82,83,84,85] and genetics [62,86,87,88,89] to characterize honey bee colonies.

The biochemistry method is not widely used for the identification of honey bees. However, Kirpik et al. (2010) [90] distinguished between *A. m. caucasia* and *A. m. remipes* with the use of a gel electrophoresis method of assessing total protein content. Russian scientists used data on enzyme activity to define the ability of honey bees to survive overwintering [35,90].

Researchers and beekeepers conduct the identification of honey bees by behavioral traits (aggressiveness, overwintering and swarming) and productivity traits at the colony level [91,92,93]. Aggressiveness has been defined by Russian beekeepers according to the reaction of the honey bees to the opening of the hive. Furthermore, it has been tested by determining positions of the bees on the honeycomb during inspection. Overwintering is the most important period for the survival and development of honey bees [94]. Swarming is a complex and labor-intensive processes occurring in apiaries which leads to the loss of honey bee colonies [95]. Beekeeping and management depend on the tendency of honey bee colonies to swarm. Swarming is one of the traits used in the description of new types or subspecies of honey bees. Many behavioral traits (swarming, aggressiveness, and others) are used to characterize colonies of honey bees [96], and this is conveniently applied by beekeepers. The best productivity of honey reported in the Russian Federation districts was presented in the Far East (Figure 3A). However, the number of honey bee colonies in the Far East was not great. Therefore, when there are fewer colonies in an environment that offers resources for bees, a smaller number of colonies may gather relatively more nectar and therefore produce more honey (Figure 3B). Hence, the performance of honey bees in the Far East was excellent.

The morphometric method was developed by a Russian scientist, Alpatov (1929) [97] Morphometry is a quantitative phenotypic method that analyzes the size of morphological traits at the individual level in honey bees [98]. Ruttner (1988) [99] evaluated bees using thirty-three morphometrics traits, although Russian scientists preferred to measure no more than eight morphometric traits in honey bees: body color, length of the proboscis, width of tergite 3, length of veins “a” and “b” of the right forewing to calculate the cubital index, width and length of the metatarsus to calculate the tarsal index, and the discoidal shift of the right forewing [42,84,100]. The morphometric traits of local and new breed types of *A. mellifera* L. honey bees were analyzed for worker bees, queens and drones. However, the number of traits to be measured has not been strictly defined by scientists. Morphometric methods have nevertheless been used to distinguish between subspecies and hybrid forms of *A. mellifera* L. [1,85,101,102]. These morphometric traits were chosen for several reasons based on the long-term study of bees [36]. The length of the proboscis was the first trait measured to differentiate between honey bees [97]. The long proboscis was usually used to define the ability of honey bees to collect nectar from deep flowers. Later, traits from different parts of the body were used to characterize honey bees: tarsal index, cubital index, length of tergite 3 and the length and width of the forewing [100]. Furthermore, these traits have been used to differentiate the subspecies *A. m. mellifera, A. m. carnica*, and *A. m. caucasia* [6,102]. Although the measurement of morphometric traits has been used to separate the subspecies of honey bees, it was not capable of distinguishing between breeds of honey bees [103]. For this reason, the variability of a traits was compared. To accomplish this, the absolute value of the coefficient of variation and the standard deviation were calculated [6,84,103,104,105,106]. It was also reported that the length and width of the forewing and body characteristics were associated with honey production [78,107,108,109]. Two morphometric traits (the length of the proboscis and length of tergite 3) and two indices (cubital and tarsal) of worker bees are presented in Table 1 and used below by us (Figure 4).

Worker bees are very well split into subspecies by means of hierarchical clustering (Figure 4B). Furthermore, the *A. mellifera* Far Eastern breed (Russian-bred) worker bees were placed into one large branch along with initial *A. m. caucasia* honey bees by means of principal component analysis (PCA) (Figure 4A). This finding provides morphometric evidence of the relationship between the *A. mellifera* Far Eastern breed (Russian-bred) honey bee and the *A. m. caucasia* honey bee. The *A. m. mellifera priokskiy* honey bee was separated from other honey bee types in *A. m. mellifera* (Figure 4A). Furthermore, this represents morphometric evidence to contrast the bred features of *A. m. mellifera priokskiy* honey bees from initial *A. m. mellifera* honey bees by locating the honey bees in different large branches of the hierarchical cluster.

The *A. m*. Far Eastern breed (Russian-bred) honey bees and the original subspecies, the *A. m. caucasia* and *A. m. mellifera* bees, were located in different areas on the PCA plot (Figure 4A,C,E). Thus, based on morphometric traits, *A. m.* Far Eastern breed (Russian-bred) workers, queen bees, and drones were separated from the original subspecies by PCA analysis. However, the hierarchical clustering method revealed another feature. The origin of the *A. m.* Far Eastern breed (Russian-bred) honey bee from the initial honey bee *A. m. caucasia* can be observed based on the location of the bees on the same branch of the dendrogram (Figure 4D,F). This feature was present in the analysis of the drones and queens, but not in that of the worker bees. It provides morphometric confirmation of the genetic relationship between *A. mellifera* Far Eastern breed (Russian-bred) honey bee and the *A. m. caucasia* honey bee.

Consequently, morphometric traits and indices of honey bees have been evaluated in assessing the differences between new types of honey bees. These morphometric traits influence the nectar collection ability [108,113], disease resistance [106], and productivity [107,114] of honey bees.

In recent years, morphometric, behavioral and productivity approaches have been utilized for the identification of honey bees in the Russian Federation [74,115,116,117]. However, rapid morphometric screening of honey bees is impossible. Thus, Russian researchers have combined this method of study with genetic methods to characterize the breeds of honey bees [89,118,119,120].

Researchers carry out genetic research on honey bees through intergenic locus polymorphism, COI-COII mitochondrial (mt) DNA, and eight microsatellite loci of nuclear DNA [61,89]. The first genetic markers of the mtDNA are unique for the *Apis* genus, and these are the most informative for bee studies [33]. This analysis allows the study of the origin of honey bees [88,121]. Furthermore, some studies based on allozyme and mtDNA variations support the findings of Ruttner (1988) [72,99,122,123,124,125].

Recently, the COLOSS Beebook was published, describing several methods for bee research, in three volumes [24], which may be useful for different research purposes. However, identification methods have not yet been generalized in the description of honey bees for all researchers.

## 3. *Apis mellifera* L. in the Russian Federation

*A. mellifera* L. honey bees are native to many places throughout the world: Europe, Africa, Eurasia, and Asia [5,121,126,127]. According to the Russian Federal State Statistics Service, in 2016 the total number of honey bee colonies was 3.3 million. Most of these (94%) exist in private apiaries in Russian territory. The Central Russian (*A. m. mellifera* L.), Carpathian (*A. m. carpatica* Avet.), Grey Mountain Caucasian (*A. m. caucasia* Gorb.), *A. m*. Far Eastern breed (Russian-bred), Bashkirsky breeds, and breed types of *A. mellifera* L. were included to the State register of Russian bees approved for breeding (http://reestr.gossort.com/reestr/animal/710, accessed on 21 June 2021). The recommendation to breed their breed types was made by scientists [11,60,111]. These honey bees have been stocked in breeding apiaries (Figure 2) and national parks. The morphometric traits of honey bees are listed in Table 1.

Moreover, investigations of *A. mellifera* Far Eastern breed (Russian-bred) honey bees have been conducted by scientists of the Research Institute of Beekeeping. They compared the *Varroa* resistance in these honey bees with local and inbred types of honey bees, and other subspecies. However, *Varroa* resistance was observed in most honey bees in Russia [6].

### 3.1. Apis mellifera mellifera L.

*A. m. mellifera* Linnaeus 1785 is commonly known as the dark European bee, and referred to as the dark bee or the Central Russian bee [115]. This honey bee is selected in breeding apiaries in seven regions of the western part of Russia and is delivered to private apiaries. Therefore, these bees constitute about 60% of the total number of honey bee colonies in the country [86].

The *A. m. mellifera* L. subspecies of the honey bee is uniquely adapted for extremely cold and long winters (lasting 6–7 months) and diseases (e.g., *Nosema* disease), as well as for the yearly harvest of honey during the short period of the rapid blossoming of the linden [115,128]. They have large body sizes, with a dark color, short proboscis, high cubital index and an average tarsal index. The body weight of these bees is 110 mg. They are distinguished by a light cup of honey cells, aggressiveness and considerable swarming behavior [11,105].

### 3.2. Breeds and Breed Types of Apis mellifera mellifera L.

On the basis of *A. m. mellifera* honey bees, a new breed (Bashkirsky), and breed types (Prioksky, Orlovsky, Tatarsky, Burzyansky bortevaya, and Prikamsky) of honey bees have been bred and certified. These bees are characterized by increased productive qualities [6,11,86].

The Bashkirsky honey bee was bred from the local population of *A. m. mellifera* L. They are calm and swarm less, displaying good overwintering and resistance to nosematose (*Nosema apis, Nosema ceranae*) compared with the initial population. These honey bees differ from their original population by rearing broods up to 10–15%, and high honey productivity up to 15% [117].

The honey bees of the Prioksky breeding type were bred in the Research Institute of Beekeeping by crossbreeding *A. m. mellifera* L. and *A. m. caucasia* Avet. breeds. The overwintering performance and resistance to diseases displayed by these bees are comparable to the levels shown by *A. m. mellifera*. These bees’ development is about 15% in spring and their swarm size is less than two times that of honey bees of *A. m. mellifera* [11]. Honey bees of this type collect the nectar from red clover flowers.

The Orlovsky breed type of honey bee was selected by researchers in the Orlovskoe Polesye apiary. The crossbreed consisted of 11 different populations of *A. m. mellifera* L. imported from natural lands. They have a high overwintering ability and are resistant to diseases. The intense spring development of the honey bee colony begins later for this type. These bees forage mostly on linden, buckwheat and cypress.

The Tatarsky breed type was bred from a local population of *A. m. mellifera*. They have the best indicators of overwintering, are calm, display less swarming and have higher development in spring than the initial honey bees. These honey bees harvest different types of honey (linden, buckwheat, and canola in particular) [11].

The Burzyansky bortevaya breed type was bred from local honey bees (*A. m. mellifera)*. They are more aggressive than the original population. These honey bees are distinguished by late but intense spring development and the queen’s great ability to lay eggs. Researchers use a “strong late-year honey harvest” from linden to describe their mass blooming performance [11,129].

The Prikamskaya breed type was developed in the Perm region [56,84]. These honey bees survive during seven months of overwintering and show resistance to *Nosema* infection. During high flowering seasons, they are able to produce an average of 21 kg of honey in one colony per day [56].

### 3.3. Apis mellifera carpatica Avet.

Breeding apiaries are also present in the southern part of the Russian Federation. These bees originated in the Carpathian Mountains where the weather is characterized by a sudden change in temperature during winter and spring. These bees are smaller, with a longer proboscis than *A. m. mellifera*. Carpathian honey bees are less aggressive during hive inspections [11]. They swarm less and the queen may lay about 1300–1800 eggs per day. These honey bees use different types of blooming plants [82].

### 3.4. Breed Types of Apis mellifera carpatica

The honey bee breed types Maikopsky and Moscovsky were bred on the basis of *A. m. carpatica*.

Maikopsky breeds are more resistant to diseases than the initial population. They are distinguished by their early spring development [112].

### 3.5. Apis mellifera caucasia Gorb.

Gray mountain Caucasian bees (*A. m. caucasia* Gorb.) are the oldest members of *A. mellifera* L. These honey bees have a gray color, weigh about 90 mg, with a long proboscis, a cubital index of 50%–55%, a negative discoidal shift and have the shape of a wax mirror in the fifth sternite [104]. They are peaceful during hive inspections and their honey cap is dark. The overwintering performance and disease resistance of *A. m. caucasia* are lower than those of *A. m. mellifera*. These honey bees are distinguished by their early spring development. The ability of the queen to lay eggs is low and does not exceed 1500 eggs per day [11]. Gray mountain Caucasian bees are extremely effective in using weak intermittent honey harvests.

### 3.6. Breed Types of Apis mellifera caucasia Gorb.

The Krasnopolyansky breeding type was bred from different populations of honey bees in the Caucasian Mountains in Russia. These bees effectively pollinate legumes, clover and alfalfa.

## 4. Honey Bees vs. *Varroa* and Tracheal Mites

*Varroa destructor* is not lethal to *A. cerana* due to host–parasite coevolution [130,131]. The reproduction of the parasite is limited to the drone brood, which restricts the population growth of the mite [132]. In contrast, in *A. mellifera* (a new host), the ability of the parasite to use both the drone brood and the more persistent worker brood leads to high rates of infestations [133,134]. The invasive parasitic mites (*Varroa destructor* and *Acarapis woodi)* are the main drivers of honey bee (*A. mellifera* L.) colony losses in beekeeping, especially in the United States [46,49]. Beekeepers have significantly relied on pesticides to control the mites, and as a consequence the chemical contamination of honey has been recorded in hives [26]. Moreover, it has been reported that mites are becoming resistant to pesticides, lessening the effectiveness of bee treatment [135]. Colony losses have been reported in Europe [136], the USA [137], the Middle East [9] and Japan [8], but not in South America, Africa or Australia [42]. It is not entirely accurate that there have been no bee losses due to *Varroa* in South America, although Africanized bees, which are present in many areas, are more resistant to the parasite [138]. In South Africa, resistance to *Varroa* also varies by subspecies [139], and in Uruguay it depends on the genetic variants of the honey bee population and the mite population [140]. This global scenario indicates the central role of this particular ectoparasitic mite in colony losses [4,19,49]. Moreover, some stocks of bees have an inherent genetic resistance to being infested by mites [141]. To determine this, Hawkins and Martin (2021) [142] studied targeted recapping behavior as an indicator of mite resistance. In the Russian Federation, *A. mellifera* Far East honey bees that are resistant to *Varroa* (*Varroa destructor*) and tracheal mites (*Acarapis woodi*) have been found in Primorsky province on the Pacific coast. This province is located in the southern part of the Far East region of Russia within the Amur River basin [143]. This *Varroa*-resistant honey bee provided the initial colonies for the breeding of new types of honey bees by Russian and American breeders. Researchers from the United States Department of Agriculture (USDA) selectively bred queens (“Russia honey bee”, American-bred) from *A. mellifera* Far East honey bees in the USA [44,144,145]. At the same time, researchers from the Federal Research Institute of Agriculture used the pure-bred breeding method to breed the *A. mellifera* Far Eastern breed (Russian-bred) honey bee in Primorsky province [50,120,146,147,148]. The Russian honey bee (American-bred) and *A. m.* Far Eastern (Chinese-bred) honey bees are used in inbred honey bee colonies in foreign conditions in the USA, Canada and China, whereas the *A. mellifera* Far Eastern breed (Russian-bred) honey bee is used in the local conditions of Russia. Various honey bee breed types are studied by Russian and North American researchers, yet the characterization methods of these bees are not identical. As a result, a partial comparison of these bee types should be provided.

### 4.1. Apis mellifera Far Eastern Breed (Russian-Bred) Honey Bees

The first apiary in the Far East of the Russian Federation was organized in 1870–1880 [58]. The variety of bee breeds that were brought to the region at the beginning of the 20th century formed a special natural mixed breed, the *A. mellifera* Far East honey bee. This initial population, used for breeding new breeds of *A. mellifera* Far Eastern breed (Russian-bred) honey bees, were obtained by means of natural interbreeding amongst bees of different subspecies (*A. m. mellifera* L., *A. m. artemisia* Engel., *A. m. caucasia* Gorb., and *A. m. ligustica* Spin.). Kodes, Popova (2016) [146] reported significant differences in morphometric traits between inbred *A. mellifera* Far Eastern breed (Russian-bred) honey bees and the initial honey bees. These breeds evolved to incorporate the best traits of the *A. mellifera* Far East bees adjusted to the local conditions [147].

Researchers at the Federal Research Institute of Agriculture, using purebred methods, bred the *A. mellifera* Far Eastern breed (Russian-bred) honey bee breed from *A. mellifera* Far East honey bees in Primorsky province in the Russian Federation according to FNC Agrobiologists of the Far East by A.K. Chaika. The new breed of honey bee consists of 6% from total honey bee colonies in the Russian Federation [6,68]. These bees are more peaceful than the *A. m. mellifera* L. honey bees. They display a strong ability of overwintering, lower swarm conditions, and higher disease resistance compared to *A. m. mellifera* L. honey bees [50]. The spring development of these honey bee colonies begins early, depending on the climate. The ability of the queen to lay eggs during the build-up to the honey harvest is 1400–1700 eggs per day. These honey bees use the last honey harvested from lime plants, bringing up to 30 kg of nectar per colony per day [6,11]. The number of honey bee colonies of *A. mellifera* Far Eastern breed (Russian-bred) honey bees is about 5500. The average honey yield was found to be 70–90 kg in *A. mellifera* Far Eastern breed (Russian-bred) honey bee colonies [147]. These honey bees have a gray and yellow color, a cubital index of 42.1%–45.4%, a weight of 107 mg, a positive discoidal shift of the wing vein, and a curved shape of the back of the wax mirror of the fifth sternite. The main morphometric traits of these honey bees exhibit some variability. Their least variable characteristics are the length of the proboscis (1.4%–3.2%) and the width of tergit 3 (1.3%–3.0%), and the most changeable feature is the cubital index (2.7%–14.0%) (Table 1).

In addition, their morphometric traits were measured and analyzed to characterize the queens and drones [6,11]. In general, *A. mellifera* Far Eastern breed (Russian-bred) honey bees are recommended for breeding in the regions of the Far Eastern Federal District in Russia. The use of other honey bee breeds in this region is not allowed [147].

### 4.2. Apis mellifera Far Eastern (American-Bred, Chinese-Bred) Honey Bees

The first study on the *A. mellifera* Far East honey bee was carried out by researchers at the United States Department of Agriculture (USDA) on the newly-bred “Russian honey bee” (American-bred). In June 1995, a test apiary was established in the Primorsky region of the Russian Federation [45,144]. Queens bees from 16 separate beekeepers were collected and brought to the Honey Bee Quarantine Station on Grand Terre Island, Louisiana, in 1997. In 1998, daughters were raised from the queens and mated with drones from the Russian queens. An evaluation of 2000 Russian honey bee colonies and 1500 domestic colonies was carried out during a harsh winter. Of the 1500 domestic colonies, 1200–1400 were lost, whereas of the 2000 Russian-bred colonies, only two were lost. The Ontario Beekeepers’ Association (OBA) in Canada (in 1999), as well as the Eastern Apiculture Society, the Saskatchewan Beekeepers’ Association (SBA), and the Canadian Honey Council (in 2001) decided to import the Russian honey bee stock for an initial evaluation as a source of *Varroa* resistance. *A. m.* Far East honey bees were introduced into Northeast China in 2009 [52]. Subsequent studies assessed the performance of inbred honey bees. The hygienic behavior was increased in inbred Russian honey bees (American-bred) in comparison with the domestic North American honey bees [53]. Furthermore, the characteristics of Russian honey bees (American-bred) was performed, marking the differences between inbred Russian and Italian honey bees [149]. Eight traits of Russian honey bees (American-bred) were assessed and compared in relation to *A. m. ligustica*. Only six traits (high resistance to *Varroa destructor,* resistance to *Acarapis woodi* tracheal mites, brood rearing during times of pollen presence in nature, low robbing ability, presence of queen cells in hives during the period of colony development, and the dark color of the honey bee body) distinguished the Russian honey bees (American-bred) [51]. However, two traits (pollination skills and a gentle temperament similar to that of *A. m. ligustica*) did not distinguish the Russian honey bees (American-bred). In addition, Russian honey bees (American-bred) developed quickly from small colonies in spring through a rapid build-up when resources were available in the field and through slowing down brood rearing when resources were lacking. Unger and Guzman-nova (2010) [51] examined the phenotypic and genotypic variabilities of the inheritance of behavior in Russian and Ontario hybrid honey bees in relation to hygienic tasks. The Russian and hybrid bees from the Russian queen bee showed a significantly higher uncapping frequency per individual than the Ontario and hybrid bees from the Ontario queen bee.

Researchers from the Russian Bee Breeders Association (RBBA) carried out research to certify the stock of Russian honey bees (American-bred) based on microsatellite and single-nucleotide polymorphism (SNP) markers [43]. As a consequence, Russian honey bees (American-bred) contributed to the gene pool of honey bees as a valuable resource for beekeeping in the United States, and provided further steps towards improving beekeeping development programs [54]. Researchers from Northeast China analyzed the biological characteristics of the introduced and inbred *A. mellifera* L. honey bees (Chinese bred), including morphometric indicators and energy metabolism indicators, as well as DNA markers [52].

## 5. Future Prospects for Beekeeping in the Russian Federation

The development of modern methodologies for breeding bees, and the evaluation of bee colonies for monitoring the gene pool of honey bees in the Russian Federation.The development of tools for the artificial insemination of queen bees and cryopreservation of drone sperm to control the pure breeding and interbreeding of honey bees.The establishment of a cryogenic gene bank of local populations of honey bees for breeding in the Russian Federation.The conservation of original types and the development of new competitor types of honey bees, considering the needs of commercial beekeeping.The breeding of queen bees with highly productive crossbreeding lines for beekeeping commodities.The development of nomadic beekeeping regulations to prevent uncontrolled mating on apiaries in the area.

## 6. Conclusions

Focusing on the prevention of honey bee losses, we identified the following steps in the recovery process: (1) find a breeding apiary with a certified stock of honey bees of a suitable subspecies; (2) breed certified honey bees under normal conditions and (3) exclude honey bees from random subspecies before breeding in the area. Thus, the recovery of lost bees was possible using honey bees with stable traits developed by means of special breeding in certain conditions over several years.

The initial *A. mellifera L.* colonies and inbred types of honey bees in the Russian Federation represent a valuable genetic stock. Natural and protected areas are critical for the preservation of *A. mellifera* honey bee populations to protect them from the uncontrolled introgression of imported strains. The application of genetic stock identification methods and regular colony evaluations are of the utmost importance for apiculture research in the Russian Federation. To supplement these certification processes, researchers could verify the geographical origins of “pure” queens and selected honey bee stocks. These molecular and morphological control techniques have great potential for the identification of honey bees. Furthermore, regional selection, evaluation, and breeding efforts for rearing queens from local honey bee populations after long-term comparative testing in their native region are essential.

## Figures and Tables

**Figure 1 insects-12-00684-f001:**
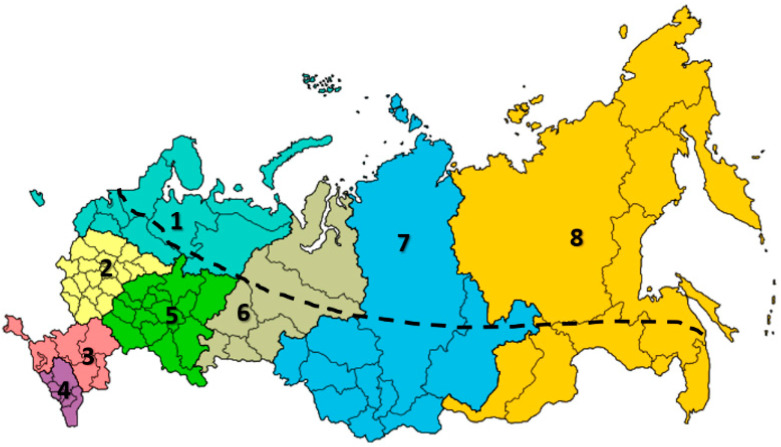
Federal districts in the Russian Federation. 1, North Western 9.87% (light blue); 2, Central 3.8% (yellow); 3, South 2.62% (pink); 4, North Caucasian 1.51% (purple); 5, Volga 6.07% (green); 6, Ural 10.62% (gray); 7, Siberian 25.50% (dark blue); 8, Far East 42.00% (orange). Border line: above the line is the northern zone of the Russian Federation. Honey bees are not present in the northern zone of the Ural district [56].

**Figure 2 insects-12-00684-f002:**
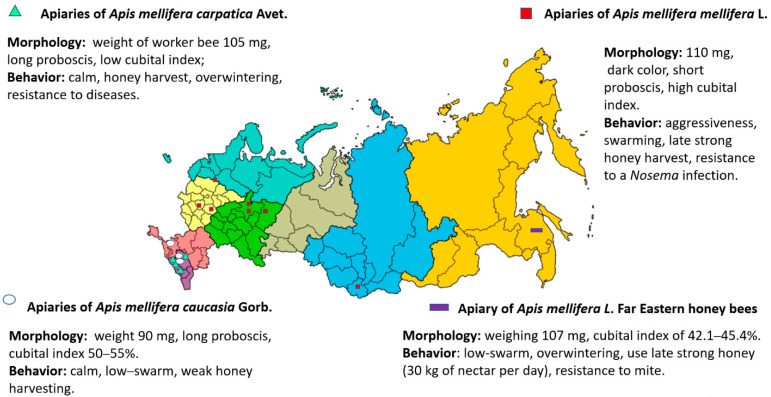
The main breeds of honey bees, locations of certified breeding apiaries, and national parks in the Russian Federation [6,11]. 1, National Park Orlovskiy Polesie (Orlov region); 2, Krasnopolyanskya Experimental Station for Beekeeping (Krasnodarskiy province); 3, ”Maikopskiy” bee-breeding apiary (Republic of Adygea); 4, Selection center “Tatarskiy” (Republic of Tatarstan); 5, Shulgan-Tash State Nature Reserve (Republic of Bashkortostan); 6, Vishersky and Malinovii Hutor national parks (Perm region).

**Figure 3 insects-12-00684-f003:**
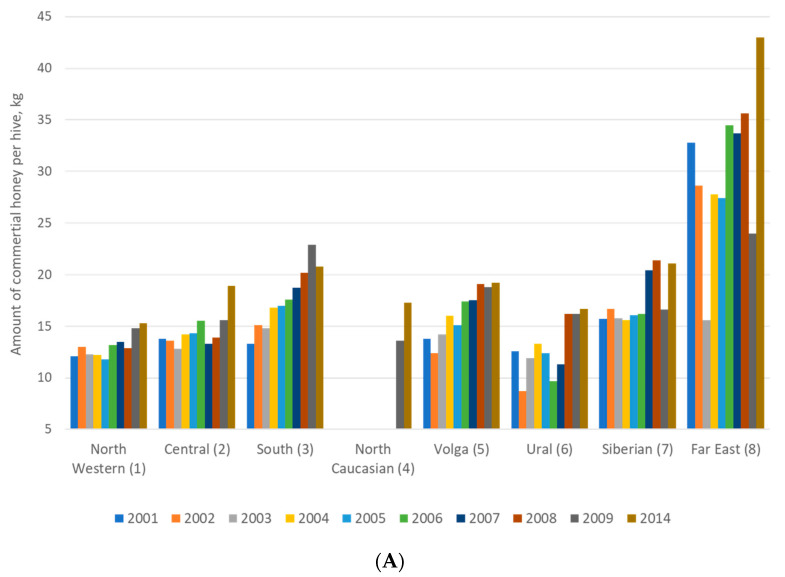
Honey productivity (**A**) and number of bee colonies (**B**) in eight federal districts in the Russian Federation.

**Figure 4 insects-12-00684-f004:**
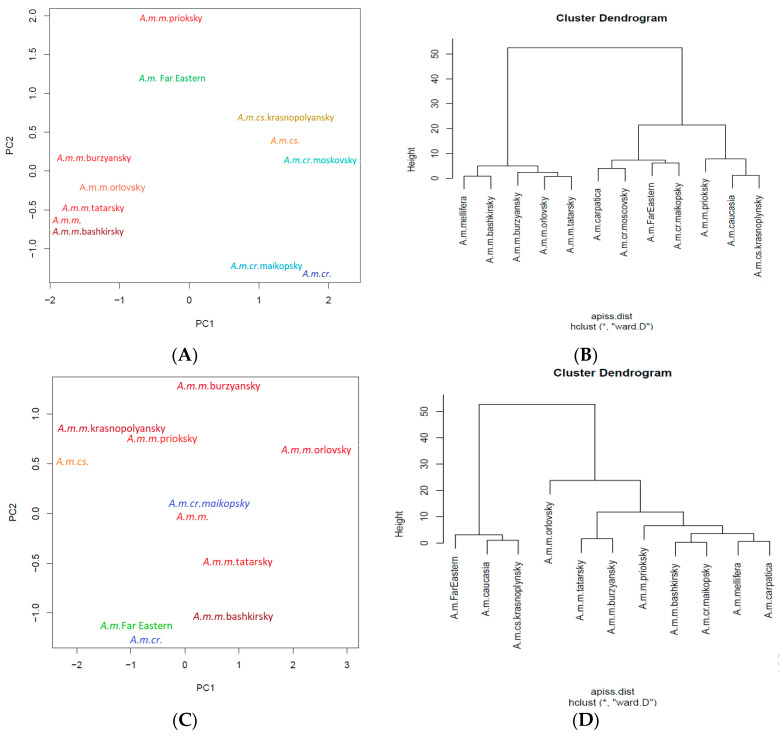
The classic morphometric discrimination between worker bees (**A**,**B**), queen bees (**C**,**D**), drones (**E**,**F**), and 12 types and subspecies of *Apis mellifera L.* honey bee colonies using PCA (**A**,**C**,**E**) and Ward’s method of hierarchical cluster analysis (**B**,**D**,**F**). A. m. m. = *Apis mellifera mellifera* honey bee; A. m. cr. = *Apis mellifera carpatica* honey bee; A. m. cs. = *Apis mellifera caucasia* honey bee.

**Table 1 insects-12-00684-t001:** Morphometric characteristics of *A. mellifera* L. worker bees from four breeds and eight bee types.

Breed, Type	Length of Proboscis, mm M ± SE	Width of Third Tergite, mm M ± SE	Cubital Index, % M ± SE	Tarsal Index, % M ± SE	Author
1. *A. m. mellifera*	6.20 ± 0.02	5.0 ± 0.04	62.3 ± 1.5	55.6 ± 0.2	[11]
1.1 Prioksky	6.70 ± 0.03	4.8 ± 0.01	56.4 ± 1.0	59.4 ± 0.3	[11]
1.2 Orlovsky	6.30 ± 0.04	4.9 ± 0.06	60.2 ± 1.7	55.8 ± 0.6	[11]
1.3 Tatarsky	6.30 ± 0.04	5.0 ± 0.01	60.6 ± 0.4	55.2 ± 0.2	[11]
1.4 Burzyansky	6.20 ± 0.03 5.85 ± 0.01	4.9 ± 0.01 4.9 ± 0.04	59.2 ± 0.5 58.4 ± 0.7	57.0 ± 0.2 55.6 ± 0.3	[11,110]
1.5 Bashkirsky	6.15 ± 0.01	5.0 ± 0.01	63.0 ± 0.2	55.1 ± 0.1	[11]
2. *A.m*. Far Eastern breed (Russian-bred)	6.70 ± 0.03 6.40 ± 0.08	5.1 ± 0.03 4.9 ± 0.01	45.4 ± 0.5 43.9 ± 0.39	57.7 ± 2.3 56.8 ± 0.2	[11,111]
3. *A.m. carpatica*	6.60 ± 0.02 6.57 ± 0.02	4.7 ± 0.01 4.6 ± 0.02	43.1 ± 0.4 39.7 ± 1.9	52.0 ± 0.6 52.6 ± 0.3	[11,112]
3.1 Maikopsky	6.70 ± 0.02	4.9 ± 0.01	47.9 ± 0.02	52.0 ± 0.1	[11]
3.2 Moscovsky	6.70 ± 0.01 6.84 ± 0.01	4.6 ± 0.01 4.6 ± 0.01	40.3 ± 1.4 38.5 ± 1.2	54.9 ± 0.2 52.5 ± 1.0	[11,112]
4. *A.m. caucasia*	6.90 ± 0.01	4.7 ± 0.01	51.2 ± 0.2	55.0 ± 0.2	[11]
4.1 Krasnopolyansky	7.00 ± 0.01	4.8 ± 0.01	52.4 ± 0.2	55.4 ± 0.3	[11]

M ± SE: M = average, SE = standard error of the mean. 1, 2, 3, 4=breed of honey bee. 1.1–1.5, 3.1–3.2, 4.1 = bee type bred from breeds 1, 3 or 4. of honey bees, respectively.

## Data Availability

Data is contained within the article.

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
