# Peer review of "Beekeeping Genetic Resources and Retrieval of Honey Bee Apis mellifera L. Stock in the Russian Federation: A Review"

_insects, 2021, doi:10.3390/insects12080684_

Round 1

Reviewer 1 Report

The review is very interesting and enhances the literature referred to the topics of Russian honey bees and their use in other countries.
The manuscript shows an in-depth overview of the situation of the characteristic honey bees of the Far East of the Russia in the past exported in the Nearctic region and then in China because of their resistance to low temperatures and parasites such as Varroa destructor and Acarapis woodi, and also because of their ability to intensively exploit the available honey resources in the short period useful for foraging in their territories of origin. Another interesting trait is their ability to intensively exploit the available honey resources in the short period needed for foraging in their territories of origin.
Examination of Russian honey bees is also important because it gives an insight into the situations reported in articles in Russian language, not easy to read.
The report highlights the current need to conserve pure strains of these honey bees, taking measures to safeguard their genetic heritage by avoiding cross-breeding, in accordance with current biodiversity conservation policies.

  • References: Authors have to follow the rules of the Journal indicated in Instructions for Authors
    - page 2, line 17, after Nosema (8%), you could add these sentences to improve the text:
    In the last years, the explanation for the decline of honey bees seems to be increasingly attributable, essentially to two causes: Phorid parasitoids and climate change. Diptera Phoridae can cause reduction or disappearance of honey bees: a damage due to Apocephalus borealis in the United States (Core et al. 2012) and to Megaselia spp. in Europe, Africa (Menail et al., 2016; Cham et al., 2018), and Asia (Debnath and Roy, 2019). The honey bee behaviour reported in Russia by Borodachev et al. (2019) could be also attributed to these parasitoids.
    References to add:
    Cham, D.T.; Fombong, A.T.; Ndegwa, P.N.; Irungu, L.W.; Nguku, E.; Raina, S.K. Megaselia scalaris (Diptera: Phoridae), an opportunist parasitoid of honey bees in Cameroon. African Entomology 2018, 26, 254–258.
    Core, A.; Runckel, C.; Ivers, J.; Quock, C.; Siapno, T.; Denault, S.; Brown, B.; Derisi, J.; Smith, C.D.; Hafernik, J. A new threat to honey bees, the parasitic phorid fly Apocephalus borealis. PLoS ONE 2012, 7, e29639, DOI:10.1371/journal.pone.0029639 Debnath, P.; Roy, D. First Record of Megaselia scalaris (Loew) as a potential facultative parasitoid of Apis mellifera in India. Asian Journal of Biology 2019, 7, 1-9; https://doi.org/10.9734/AJOB/2018/46210
    Dutto, M.; Ferrazzi, P. Megaselia rufipes (Diptera: Phoridae): a new cause of facultative parasitoidism in Apis mellifera. Journal of Apicultural Research 2014, 53, 141-145; DOI: 10.3896/IBRA.1.53.1.15.
    Menail, A.H.; Piot, N.; Meeus, I.; Smagghe, G.; Loucif-Ayad, W. Large pathogen screening reveals first report of Megaselia scalaris (Diptera: Phoridae) parasitizing Apis mellifera intermissa (Hymenoptera: Apidae). Journal of Invertebrate Pathology 2016, 137, 33-37.
    Ricchiuti, L.; Miranda, M.; Venti, R.; Bosi, F.; Marino, L.; Mutinelli F. Infestation of Apis mellifera colonies by Megaselia scalaris (Loew, 1866) in Abruzzo and Molise regions, central-southern Italy, Journal of Apicultural Research. 2016, 55, 187-192; DOI: 10.1080/00218839.2016.1196017
    - page 2, line 26, after (Shestakova, 2021) insert:
    Climate change threats honey bees in terms of lack or reduction of resource (nectar, pollen and honeydew) and spread of Varroa (Vercelli et al., 2021) Vercelli, M.; Novelli, S.; Ferrazzi, P.; Lentini, G.; Ferracini, C. A Qualitative Analysis of Beekeepers' Perceptions and Farm Management Adaptations to the Impact of Climate Change on Honey Bees. Insects 2021 12, 228; doi: 10.3390/insects12030228.
    page 2, line 53 tracheal
    - page 4, line 24
    It is not entirely accurate that there have been no bee losses due to varroa in South America, although Africanized bees, which are present in many areas, are more resistant to the parasite (Maggi et al. 2016). In South Africa, resistance to Varroa also varies by subspecies (Martin & Kriger 2001).
    References to add:
    Maggi, M.; Antunez, K.; Invernizzi, C.; Aldea, P.; Vargas, M.; Negri, P., Brasesco, C.; De Jong, D.; Message, D.;, Weinstein Teixeira, E.; Principal, J.; Barrios, C., Ruffinengo, S.; Rodriguez Da Silva, R.; Eguaras, M. Honeybee Health in South america. Apidologie 2016, 835-854.
    Martin, S.J.; Kriger, P. Impact of Varroa destructor on the honey bees of South Africa. Abstracts 37th International Congress , Durban, South Africa 2001, 47-48.
    - page 4, line 33
    You can cite papers about the Russian bred resistant to Varroa and tracheal mites, for example De Guzman et al. 2001 American Bee Journal...
    De Guzman, L.I.; Rinderer, T.E.; Delatte, G.T.; Stelze,J.A.; Beaman, L.; Harper, C. An evaluation of Far-eastern Russian Honey Bees and Other Methods for the Control of Tracheal Mites. Apicultural Research 2001, 737-741.
    - page 5, line 15 tracheal: small
    - page 5 lines 41-42 Apis mellifera acervorum , new name Apis mellifera artemisia Engel 1999 (Ilyasov et al. 2020)
    Ilyasov, R.A.; Lee, M-I.; Takahashi, J-i.; Wook Kwon, H.; Nikolenko, A.G. A revision of subspecies structure of western honey bee Apis mellifera. Saudi Journal of Biological Sciences 2020, 27, 3615-3621; https://doi.org/10.1016/j.sjbs.2020.08.001
    - page 6, line 20 in all Russian territory instead “in the all lands of Russian”
    - page 7, Fig. 2 Apis mellifera L.
    - page 7, line 24 Nosema ceranae
    - page 8, line 15 They use strong late-year honey harvested from linden
    What does it means about “strong”? - page 11, Fig.3 Caption. Delete ; after Federation - page 11, line 8: Delete . after mellifera
    - page 11, line 18: to define
    - page 11, line 20: Additionally, variability in season was marked for the length of proboscis. What does it mean?
    - page 11, line 26-28: Add in the sentence “At the same time, the A. mellifera Far Eastern worker bees were placed in one big branch with initial A. m. caucasia honey bees by Principal Component Analysis (PCA)
    -page 11, line 30-31-32: A. m. mellifera prioksky
    - page 13, lines 3-5 after caption: “But relationships this honey bees observed by hierarhial clus-tering method”
    Explain better the sentence please
    - page 13, lines 20-24 after caption: “Researchers carry out genetic research on honey bees through intergenic locus poly-morphism COI-COII mitochondrial (mt) DNA and eight microsatellite loci of nuclear DNA (Ostroverkhova et al., 2016; Syromyatnikov et al., 2018). Similarity, genetic markers are unique for the Apis genus”
    it was meant to write Similarly?
    - References: Authors have to follow the rules of the Journal indicated in Instructions for Authors
    References ï‚· 24 Le Conte Y. Honey Bees surviving Varroa destructor infestations in the World. Lessons we can take. 2016. 2016 International Congress of Entomology, 2016 DOI... ICE
    32 Domènech, R.; Tracy, E.F.; Rovira, M., Lepeshkin, E. Beekeeping in Primorsky Province: Challenges and Opportunities. A Needs Assessment Report. Forest Science and Technology Centre of Catalonia (CTFC) and WWF Russia, 2019.
    46 Ilyasov, R. A., Poskryakov A. V., Petukhov A. V., Nikolenko A. G. 2016. Molecular ge-netic analysis of five extant reserves of black honeybee Apis mellifera mellifera
    51 Kandemir...Apidologie 2011, 42, 618-627. https....
    66 Meixner... Best Bee: An Experiment...Journal , 155, 663-666 67 Mehlim.....n.d. 2010
    75 Papachristoforou.... based on Microsatellites
    LEGENDA
    In blue my suggestions and my references suggested, in red my corrections, highlighted in yellow the parts to be removed.

Author Response

Dear Editor-in-Chief and Reviewer 1,

Thank you for the time and effort spent reviewing our manuscript and suggesting some important points to consider.

Our replied marked by red colour after symbol R (reply) and question number (1-21). Also, we give the page number and lines in the revision manuscript where these comments changed. But for convenience, we copied this text also in response.

We attached one world document. The first text 1 is a revised manuscript according to instructions and comments. Track changes present in text 2 below text 1. 

Please, find below the reviewer comments (black colour) and our responses (red colour).

Reviewers comments:

  1. References: Authors have to follow the rules of the Journal indicated in Instructions for Authors R1: We accept your suggestion and changed References according to the Instructions for Authors. Please see Pages 14 – 21.

  1. Page 2, line 17, after Nosema (8%), you could add these sentences to improve the text:
    «In the last years, the explanation for the decline of honey bees seems to be increasingly attributable, essentially to two causes: Phorid parasitoids and climate change. Diptera Phoridae can cause weakening, reduction or disappearance of honey bees: damage due to Apocephalus borealis in the United States (Core et al. 2012) and to Megaselia spp. in Europe (Ricchiuti et al, 2015), Africa (Menail et al., 2016; Cham et al., 2018), and Asia (Debnath and Roy, 2019). The first warning was reported in 2014 year, when the parasite Megaselia scalaris (Loew) was not present in Russian Federation and was introduced with fruits from Japan, Germany, Turkey and other countries (official documents of the Federal Service for Veterinary and Phytosanitary Surveillance of Russian Federation). However, Megaselia spp. presented in the native environment in Russian Federation, but their live cycle was restricted by low temperature (Chaika, 2003). The loss of honey bees because of infestation Megaselia spp. was not detected yet. The honey bee behaviour reported in Russia by Borodachev et al. (2019) could be also attributed to these parasitoids». References to add:

Cham, D.T.; Fombong, A.T.; Ndegwa, P.N.; Irungu, L.W.; Nguku, E.; Raina, S.K. Megaselia scalaris (Diptera: Phoridae), an opportunist parasitoid of honey bees in Cameroon. African Entomology 2018, 26, 254–258.

Core, A.; Runckel, C.; Ivers, J.; Quock, C.; Siapno, T.; Denault, S.; Brown, B.; Derisi, J.; Smith, C.D.; Hafernik, J. A new threat to honey bees, the parasitic phorid fly Apocephalus borealis. PLoS ONE 2012, 7, e29639, DOI:10.1371/journal.pone.0029639

Debnath, P.; Roy, D. First Record of Megaselia scalaris (Loew) as a potential facultative parasitoid of Apis mellifera in India. Asian Journal of Biology 2019, 7, 1-9; https://doi.org/10.9734/AJOB/2018/46210

Dutto, M.; Ferrazzi, P. Megaselia rufipes (Diptera: Phoridae): a new cause of facultative parasitoids in Apis mellifera. Journal of Apicultural Research 2014, 53, 141-145; DOI: 10.3896/IBRA.1.53.1.15.

Menail, A.H.; Piot, N.; Meeus, I.; Smagghe, G.; Loucif-Ayad, W. Large pathogen screening reveals the first report of Megaselia scalaris (Diptera: Phoridae) parasitizing Apis mellifera intermissa (Hymenoptera: Apidae). Journal of Invertebrate Pathology 2016, 137, 33-37.

Ricchiuti, L.; Miranda, M.; Venti, R.; Bosi, F.; Marino, L.; Mutinelli F. Infestation of Apis mellifera colonies by Megaselia scalaris (Loew, 1866) in Abruzzo and Molise regions, central-southern Italy, Journal of Apicultural Research. 2016, 55, 187-192; DOI: 10.1080/00218839.2016.1196017

R2: We thank the reviewer for this comment and agree with the importance to highlight the Megaselia spp as the reason of loss honey bees, not mentioned in the COLOSS report in 2015. We changed the part and clarified this problem in Russia. Also, references were added. Please see Page 2, lines 12 – 23: “Furthermore, the explanation for the decline of honey bees seems to be increasingly attributable not listed above in COLOSS research to Phorid parasitoids. Diptera Phoridae can cause the weakening, reduction, or disappearance of honey bees, with damage caused by Apocephalus borealis in the United States [31] and by Megaselia ssp. in Europe [112], Africa [27, 93], and Asia [39]. The first warning was reported in Russian Federation in 2014, when the parasite Megaselia scalaris (Loew) was introduced with plants [104]. However, Megaselia ssp. (not Megaselia scalaris) were present in the native environment in the Russian Federation, but their life cycle was restricted by low temperatures [26]. The mass loss of honey bees caused by infestations of Megaselia scalaris has not yet been detected in Russian Federation [110, 118, 144]. All other reasons (starvation, robbing, unspecified winter loss, other diseases, unknown reasons) accounted for 37% of the losses [92]. Similar issues exist in beekeeping in the Russian Federation and other countries [6, 20, 22, 23, 25, 28, 29, 74].”.

  1. page 2, line 26, after (Shestakova, 2021) insert:
    Climate change threatens honey bees in terms of lack or reduction of resources (nectar, pollen and honeydew) and spread of Varroa (Vercelli et al., 2021) Vercelli, M.; Novelli, S.; Ferrazzi, P.; Lentini, G.; Ferracini, C. A Qualitative Analysis of Beekeepers' Perceptions and Farm Management Adaptations to the Impact of Climate Change on Honey Bees. Insects 2021 12, 228; doi: 10.3390/insects12030228. R3: We thank the reviewer for this comment and agree with the influences of climate change on spreading new pests. We re-wrote this part according to recommendations. Please see Page 2, lines 24 – 26 “In addition, climate change threatens honey bees in terms of the lack or reduction of resources (nectar, pollen, and honeydew) and the spread of Varroa and other pests [50, 142].”

  1. page 2, line 53 R4: We accept your remark and we changed all sentences rewriting, not only one. Please see the page 1, line 31; page 11, lines 8, 30; page 12, line 50.

  1. page 4, line 24 “It is not entirely accurate that there have been no bee losses due to Varroa in South America, although Africanized bees, which are present in many areas, are more resistant to the parasite (Maggi et al. 2016). In South Africa, resistance to Varroa also varies by subspecies (Martin & Kriger 2001), and depending on genetic variants of the honey bee population and mite population (Mendoza et al, 2020)”.
    References to add:

Mendoza, Tomasco, Antunez Unraveling Honey bee – Varroa destructor Interaction 2020
Maggi, M.; Antunez, K.; Invernizzi, C.; Aldea, P.; Vargas, M.; Negri, P., Brasesco, C.; De Jong, D.; Message, D.;, Weinstein Teixeira, E.; Principal, J.; Barrios, C., Ruffinengo, S.; Rodriguez Da Silva, R.; Eguaras, M. Honeybee Health in South America. Apidologie 2016, 835-854.
Martin, S.J.; Kriger, P. Impact of Varroa destructor on the honey bees of South Africa. Abstracts 37th International Congress, Durban, South Africa 2001, 47-48.

R5: We appreciated the suggestion and this was added. Please see page 11, lines 22 – 26 It is not entirely accurate that there have been no bee losses due to Varroa in South America, although Africanized bees, which are present in many areas, are more resistant to the parasite [88]. In South Africa, resistance to Varroa also varies by subspecies [90], and in Uruguay, it depended on the genetic variants of the honey bee population and the mite population [94]”.

  1. page 4, line 33 Gusman et al. (2001) evaluated the effectiveness of woodi control treatments when combined with treatment designed for the simulations control of V. destructor. They found that the strong resistance of Primorsky honey bees to tracheal mites in itself was sufficient to minimize tracheal mite damage.
    You can cite papers about the Russian bred resistance to Varroa and tracheal mites, for example, De Guzman et al. 2001 American Bee Journal...
    De Guzman, L.I.; Rinderer, T.E.; Delatte, G.T.; Stelze,J.A.; Beaman, L.; Harper, C. An evaluation of Far-eastern Russian Honey Bees and Other Methods for the Control of Tracheal Mites. Apicultural Research 2001, 737-741. R6: We appreciated the suggestion and this was changed. Please see page 3, lines 12 – 16 “De Gusman et al. (2001) [35] evaluated the effectiveness of A. woodi control treatments in the breeding of Russian honey bees (in inbred honey bees from A. mellifera Far East honey bee). They found that the strong resistance of inbred Russian honey bees (American-bred bees) to tracheal mites was in itself sufficient to minimize tracheal mite damage”.

  1. page 5, line 15 tracheal: small R7: We accept your remark and we changed all sentences rewriting, not only one. Please see page 3, lines 15, 16, 17; page 12, line 52.

  1. page 5 lines 41-42 Apis mellifera acervorum, new name Apis mellifera artemisia Engel 1999 (Ilyasov et al. 2020)

Ilyasov, R.A.; Lee, M-I.; Takahashi, J-i.; Wook Kwon, H.; Nikolenko, A.G. A revision of subspecies structure of western honey bee Apis mellifera. Saudi Journal of Biological Sciences 2020, 27, 3615-3621; https://doi.org/10.1016/j.sjbs.2020.08.001 R8: We accept your remark and we changed this mistake by re-writing the sentence. Please see page 12, line 3. This initial population, used for breeding new breeds of A. mellifera Far Eastern breed (Russian-bred) honey bees, was obtained through natural interbreeding amongst bees of different subspecies (A. m. mellifera L., A. m. artemisia Engel., A. m. caucasia Gorb., and A. m. ligustica Spin.)”.

9. page 6, line 20 in all Russian territory instead “in all lands of Russian” R9: We appreciated the suggestion, but this phrase was not applied for the reason of rewriting some sentences.

10. page 7, 2 Apis mellifera L. R10: We appreciated the suggestion and this mistake was changed. Please see Figure 2 on page 5.

11. page 7, line 24 Nosema ceranae R11: We appreciated the suggestion and this mistake was changed. Please see page 10, line 5.

12. page 8, line 15 They use strong late-year honey harvested from linden
What does it mean about “strong”? –R12: Strong honey harvest means mass blooming during the short time in the environment and honey bees chose the plants for nectar gathering. The honey harvest restricted before and after mass blooming plants.

13. page 11, Fig. 3 Delete; after Federation R13: We appreciated the suggestion and these words were changed. Please see page 5, Figure 3.

14. page 11, line 8: Delete; after mellifera R14: We appreciated the suggestion and these words were deleted.

15. page 11, line 18: to define R15: We appreciated the suggestion and these words were changed.

16. page 11, line 20: Additionally, variability in the season was marked for the length of the proboscis. What does it mean? R16: The Russian scientists have a recommendation to collect the samples for morphometry in the autumn season because the colony condition influenced the size of worker bees through feeding. Restriction feeding of larvae (no pollen, no nectar in environment) and weak colonies lead to the lessness size of worker bees in the spring season in the north part of Russia. The lessness size was observed no all morphometric traits, but the proboscis was included (by personal communication and unpublished data with Berezin A., Petukhov A.).

17. page 11, line 26-28: Add in the sentence “At the same time, the mellifera Far Eastern worker bees were placed in one big branch with initial A. m. caucasia honey bees by Principal Component Analysis (PCA) R17: We accept your remark and we corrected the sentence. Please see page 7, lines 10 – 12. “Furthermore, the A. mellifera Far Eastern breed (Russian-bred) worker bees were placed into one large branch along with initial A. m. caucasia honey bees employing principal component analysis (PCA)”.

  1. page 11, line 30-31-32: m. mellifera prioksky R18: We accept your remark and we corrected the sentence.

  1. page 13, lines 3-5 after caption: “But relationships these honey bees observed by hierarchical clustering method”. Explain better the sentence please R19: PCA, MDS and hierarchial clustering analysis are the Data mining methods. The purpose of all of these methods estimation the numerous data of samples and finding something relations between samples which should be calculated later by statistic analyses. PCA, MDS, clustering differed by calculations of the numerous data to construct the plots. These calculations were highlighted the special morphometric peculiarities in our case. PCA visualized the morphometric separations of A. m. Far Eastern breed (Russian-bred) honey bee from initial subspecies. This PCA result supported the practice of breeding these honey bees as a breed type from initial honey bees. However, the hierarchical clustering method was applied the same morphometric data as PCA. A. m. Far Eastern breed (Russian bred) honey bee near located with initial honey bees on the branch. So, we explained this as the origin of A. m. Far Eastern breed (Russian-bred) honey bees visualized by clustering tree.
  2. page 13, lines 20-24 after caption: “Researchers carry out genetic research on honey bees through intergenic locus polymorphism COI-COII mitochondrial (mt) DNA and eight microsatellite loci of nuclear DNA (Ostroverkhova et al., 2016; Syromyatnikov et al., 2018). Similarity, genetic markers are unique for the Apis genus”it was meant to write Similarly? R20: Thank you for this remark. “Similarity” was not clear applied in this case and we corrected the sentence. Please see page 9, lines 6 - 10. Researchers carry out genetic research on honey bees through intergenic locus polymorphism, COI-COII mitochondrial (mt) DNA, and eight microsatellite loci of nuclear DNA [105, 132]. The first genetic markers of the mtDNA are unique for the Apis genus, and these are the most informative for bee studies [32]. This analysis allows the study of the origin of honey bees [130, 107]”.

  1. References: Authors have to follow the rules of the Journal indicated in Instructions for Authors
    References

24 Le Conte Y. Honey Bees surviving Varroa destructor infestations in the World. Lessons we can take. 2016. 2016 International Congress of Entomology, 2016 DOI... ICE
32 Domènech, R.; Tracy, E.F.; Rovira, M., Lepeshkin, E. Beekeeping in Primorsky Province: Challenges and Opportunities. A Needs Assessment Report. Forest Science and Technology Centre of Catalonia (CTFC) and WWF Russia, 2019.
46 Ilyasov, R. A., Poskryakov A. V., Petukhov A. V., Nikolenko A. G. 2016. Molecular genetic analysis of five extant reserves of black honeybee Apis mellifera mellifera
51 Kandemir...Apidologie 2011, 42, 618-627. https...
66 Meixner... Best Bee: An Experiment...Journal, 155, 663-666 67 Mehlim.....n.d. 2010
75 Papachristoforou.... based on Microsatellites
R21: We accept your suggestion and changed References according to the Instructions for Authors. Please see Pages 14 – 21.

Thank you for your accurate explanation of our discrepancies. This manuscript was revised according to your comments. Also, we used the English correction by professional language editing services.

We would like to express our heartfelt gratitude again for the insightful and encouraging comments from the reviewers.

Yours sincerely,

Dr Choi Y. S.

Dr Frunze O

Reviewer 2 Report

The objective of the work is very important. Authors provided substantial details regarding the value of the genetic stock of Apis mellifera in the Russian Federation with excellent relating to its "extensions" in other parts of the world. However, I have some comments throughout the manuscript that need to be clarified by authors. I also recommend English language editing to the ms.

Author Response

Dear Editor-in-Chief and Reviewer 2,

Thank you for the time and effort spent reviewing our manuscript and suggesting some important points to consider.

We attached the manuscript "Round 1...". The first text (1-21 pages) has no track changes, this text corrected by English editor and me according to Instructions for authors. The second text (22-52 pages) after (below) first text have track changes, but not corrected English editor. Sorry for this complication.

Our replied marked by red colour after symbol R (reply) and question number (1-51). For convenience, we copied this text also in response.

Please, find below the reviewer comments (black colour) and our responses (red colour).

Reviewers comments:

  1. The title of the review is not reflecting the detailed content. It also somehow misleading. "Beekeeping resources" is a broad term. The title could be better changed to fully represent the review to: Beekeeping genetic resources and retrieving of honey bee Apis mellifera stock in The Russian Federation. R1: We thank the reviewer for their comment and have changed the title of the review Beekeeping genetic resources and retrieval of honey bee Apis mellifera L. stock in the Russian Federation: A Review. “Retrieving” was changed by professional language editing services on “retrieval”. Please see Page 1.

  1. Review. R2: We accept your remark and we changed the sentence.

  1. How a review study has results? It can be a conclusion/ suggestion based on the available literature. R3: We thank the reviewer for their comment. We changed all sentences with this word.

  1. Review. R4: We accept your remark and we changed the sentence.

  1. References? R5: Yes, this information in the abstract was analyzed from literature.

  1. This is not clear! Does it mean "strong ability for overwintering"? if that's right, please correct it. R6: We accept your remark and we changed the sentence.

  1. "collect nectar" How nectar collection in high blooming is the main trait? what is the challenge here compared to nectar collection in low blooming? R7: We accept your remark and we changed on “different aptitude to nectar collection in low or high blooming season”. However, high blooming means mass blooming during the short time in the environment and honey bees chose the plants for nectar gathering. The honey harvest so much restricted before and after mass blooming plants (low blooming season) in Russia.

  1. The following keywords are suggested and may replace some of the keywords provided: Russian beekeeping, Bee breeding, Honeybee losses, Apis mellifera breeds, bee mites resistance. Check how to put keywords for scientific article. R8: We appreciated the suggestion and these words was changed in keywords.

  1. The author must pay attention to the grammar and sentence structure for easy understanding and clarity of the text. The coordination between the paragraphs can be better improved for easy understanding of the reader. The author must follow a sequence to present/ introduce the information regarding the beekeeping resources and the challenges that are being faced in beekeeping, and the ways to restore the beekeeping resources. At the end of introduction, the author should squeeze the text by highlighting the need of this review and the points that are being targeted for this review. The author stated the word "our results" which is applicable for any investigative study with real experiments and recoding of data. However, for a review study, the author may give conclusions, recommendations and the present facts (from previous literature). R9: We thank the reviewer for this comment and agree with valuable comment. We re-wrote the introduction according your recommendations.

  1. What was the outcome or conclusion of that conference regarding honey bee losses? better to be added. R10: We thank the reviewer for this valuable comment. The conclusion of that conference was added. Please see page 2, lines 29 - 44. Also, this text presents below:

“The reasons for honey bee losses and methods of assessment and protection were discussed widely at the International Scientific and Practical Conference on “Modern Problems of Beekeeping and Apitherapy” (Russia, Rybnoye, 2020), and the International Conference on “Modern aspects of apitherapy” (Ukraine, Kiev, 2020). The environmental situation has deteriorated around the world during recent years. It was noted that the crossbreeding of honey bees, chemical treatments of plants, insects, and factory emissions have played a major role in this deterioration. The accumulation of metal ions in honey bee bodies has influenced the health of honey bee populations in the Ryazan region [85]. E. K. Eskov, M. D. Eskova (2021) [43] analyzed honey bee losses in polluted areas as a result of the accumulation of pollutants in the honey bee body. Moreover, the performance of honey bees was assessed in terms of hygienic behavior, which was linked with morphometry characteristics (cubital index, tarsal index, length of proboscis, and the width of tergite 3 and wing veins), honey productivity, and Varroa infestations [11]. In addition, the effectiveness of biotechnological methods has been reported for the prevention of Varroa mites in honey bee populations, including heating and the use of mite traps with the naturally attractive properties of medicinal plants [20]”.

  1. Any beekeeper in the world has either: local, foreign, or inbred colonies! Better to modify this statement. R11: Russia is a very big country. This is specific to Russia, theirs are bad and good sides. There presents so far village with the poor citizens in Ural, Siberian. The road cover doesn’t present there. Nobody delivers anything in this village. But, natural resources are very rich there. So, traditional beekeeping is very specific there: no foreigner honey bee, no breed honey bee from breeding apiary or other countries. Beekeepers have no foreign honey bees by reason of the distance, bad road, expensive cost to move honey bees or order the purebred foreign queen. Actually, the foreign queen cages cannot quickly deliver on 2-3 thousand km. So, they breed only local honey bees by the traditional selection method year by year. Also, members from the beekeepers’ institute connect with beekeepers in Russia through communication with local beekeepers’ centres. But the south and Moscow region of Russia has no problem like this. That why local, inbred and foreign colonies have to differ in Russia.

  1. Are you talking about any specific honey bee subspecies or native subspecies only? R12: We thank the reviewer for this comment and agree that we mean the native honey bees. Please see page 2, lines 45-47. Also, this text presents below: “Currently, beekeepers breed local, foreign, or inbred honey bee colonies. Traditional beekeeping was based on the breeding of local honey bee resources in a native environment [1, 4, 67, 122]”.

  1. This is not always the case in all regions of the world. Rephrase this sentence. R13: We accept your offers and the sentences have been rewrote. Please see page 2, lines 51-54, page 3, lines 1-2. Also, this text presents below: “The traits of longevity, gentleness, and high honey production were examined in local and foreign honey bees [34, 36]. Sometimes, benefits were introduced into the local honey bee populations [16, 92]. However, new diseases were also introduced along with foreign honey bees, which has been the cause of local and foreign honey bee colony losses”.

  1. This "interrupted" very short statement needs more explanation or details. R14: We thanks the reviewer for their comments and explained more. Please see page 3, lines 3-20. Also, this text presents below: “With the exception of pure local colonies, bee breeders breed new (inbred) types from foreign or local honey bees using selection methods and comparing their traits with initial or domestic colonies [86]. Inbred mellifera Far Eastern breed (Russian-bred) honey bees were bred from A. mellifera Far East honey bees [123]. Furthermore, the latter subspecies of honey bees were also the initial colony used for the breeding of Russian honey bees (American-bred) in the USA, hybrids of Russian and Ontario honey bees in Canada [138], and Varroa-resistant A. mellifera Far Eastern honey bees in China (Chinese-bred) [146]. As a consequence, inbred honey bees in the USA, Canada, and China all originated from introduced A. mellifera Far East honey bees. Inbred honey bees have special traits in comparison with domestic colonies. De Gusman et al. (2001) [35] evaluated the effectiveness of A. woodi control treatments in the breeding of Russian honey bees (in inbred honey bees from A. mellifera Far East honey bee). They found that the strong resistance of inbred Russian honey bees (American-bred bees) to tracheal mites was in itself sufficient to minimize tracheal mite damage. Thus, the resistance to Varroa and tracheal mites displayed by the initial honey bees (A. mellifera Far East) were transmitted to Russian honey bees (American-bred) during breeding [114]. Thus, some advantages are displayed by local and inbred honey bees in comparison with foreign colonies”.

  1. Confusion: In order to do something before 2010 as reported by Guzman-novoa, bee breeders start something in 2017!! Another confusion: The start of queens’ imports from the USA to Canada started 2017 reported by Van der Zee et al., (2012). Confusion here. How come the information published in 2012 about something in 2017? Also, the reference Van der Zee et al., 2012 is not the same in the list of references. R15: We accept your remarks about this confusion and have checked and have rewritten this text. Also, all citation was checked in text and in reference. Please see page 12, lines 30-43. Also, this text presents below:

The first study on the A. mellifera Far East honey bee was carried out by researchers at the United States Department of Agriculture (USDA) on the newly-bred “Russian honey bee” (American-bred). In June 1995, a test apiary was established in the Primorsky region of the Russian Federation [37, 115]. Queens bees from 16 separate beekeepers were collected and brought to the Honey Bee Quarantine Station on Grand Terre Island, Louisiana, in 1997. In 1998, daughters were raised by the queens and mated with drones from the Russian queens. An evaluation of 2000 Russian honey bee colonies and 1500 domestic colonies was carried out during a harsh winter. Of the 1500 domestic colonies, 1200–1400 were lost, whereas, of the 2000 Russian-bred colonies, only two were lost. The Ontario Beekeepers' Association (OBA) in Canada (in 1999), as well as the Eastern Apiculture Society, the Saskatchewan Beekeepers' Association (SBA), and the Canadian Honey Council (in 2001) decided to import the Russian honey bee stock for an initial evaluation as a source of Varroa resistance. A. m. Far East honey bees were introduced into Northeast China in 2009 [146]”.

  1. Instead of the word "results", It is better to use a conclusion because the review study can give a conclusion based on the previous literature. R16: We accept your remark and we changed the sentence. This comments was corrected in all text of the manuscript.

  1. beekeeping genetic resources R17: We accept your remark and we changed the sentence.
  2. Apply the numbers (1-8) to each district in the text for comparing the facts with figure 1 (Figure- 1 only shows numbers for each district). For example, 8- Far East (42%), 7-Serberian (25.5%). (Changed text: The country has 8 federal districts: 1 – North Western (9.87 % of all Russian Federation), 2 – Central (3.80%), 3 – South (2.62%), 4 – North Caucasian (1.51%), 5 – Volga (6.07%), 6 – Ural (10.62%), 7 – Siberian (25.50%), 8 – Far East (42,00%) (Figure 1)). Add percentages in figure legend as described in the text. For example, 1-North Western 9.87% (light blue) (Changed text: Figure 1. Federal districts in Russian Federation. 1 – North Western 9.87 % (light blue), 2 – Central 3.8% (yellow), 3 – South 2.62% (pink), 4 – North Caucasian 1.51% (purple), 5 – Volga 6.07 % (green), 6 – Ural 10.62% (gray), 7 – Siberian 25.50% (dark blue), 8 – Far East 42.00% (orange). Border line: above which is the Northern zone of the Russian Federation. The honey bees are not present in the Northern zone of Ural (Murylev, Petukhov, 2014)). R18: We appreciated the suggestion and reformatted the Figure 1 and text. Please see page 3, lines 33-41. Also, this text presents below: “The country has eight federal districts: 1—North Western (9.87% of all Russian Federation), 2—Central (3.80%), 3—South (2.62%), 4—North Caucasian (1.51%), 5—Volga (6.07%), 6—Ural (10.62%), 7—Siberian (25.50%), 8—Far East (42.00%) districts (Figure 1).

Figure 1. Federal districts in the Russian Federation. 1—North Western 9.87% (light blue), 2—Central 3.8% (yellow), 3—South 2.62% (pink), 4—North Caucasian 1.51% (purple), 5—Volga 6.07% (green), 6—Ural 10.62% (gray), 7—Siberian 25.50% (dark blue), 8—Far East 42.00% (orange). Border line: above the line is the Northern zone of the Russian Federation. The honey bees are not present in the Northern zone of the Ural district [100]”.

  1. What was the correct description of the local honeybee? There is confusion here: The author stated that mellifera was 150 years old but was described as Apis cerana. Then, no confirmation was provided of the presence of A. cerana together with A. mellifera in the same region. R19: We thank the reviewer for this comment. The past information did not follow the main purpose of this review and it was removed. Please see page 4, lines 1-5. Also, this text presents below: “The largest part of the Russian Federation is the Far East, where A. mellifera L. beekeeping is less than 150 years old [80, 149]. The local forest honey bee was described by Kozhevnicov in 1926 as A. cerana (in modern taxonomy). At that time beekeepers did not breed A. cerana honey bees in apiaries [82, 60]. Both honey bees survived at average temperatures…”

Also, Ruttner studied the morphometrical differences A. mellifera and A. cerana honey bees. First times nobody differed this honey bees as separate species. F. Ruttner, L. Tassencourt, J. Louveaux. BIOMETRICAL-STATISTICAL ANALYSIS OF THE GEOGRAPHIC VARIABILITY OF APIS MELLIFERA L.* I. Material and Methods. Apidologie, Springer Verlag, 1978, 9 (4), pp.363-381. “First of all, the graph shows the clear biometrical separation of the two species A. mellifera and A. cerana. It should be noted here that the analysis employs only characters that proved to be of value for discrimination within the species A. mellifera. No character typical for A. cerana was incorporated (i.g. tomentum on tergite 5, radial vein on the hind wing, weak barbs on the sting). Otherwise, A. cerana would shift much further away from A. mellifera on the graph.”

  1. Since, the author targeted two species (Apis mellifera and Apis cerana), is the honey yield fact in this sentence valid for both species? R20: Yes, Excellent bee forage ensures significant honey yield by honey bees ( mellifera and A. cerana).

  1. what months? “In Siberia, beekeeping was introduced about 230 years ago and overwintering of honey bees lasts from 5 to 6 months (from November to April) (Ostroverkhova et al., 2016)”. R21: We thank the reviewer for their comments and explained as we see this question. It is mean, that honey bee stays inside the and cannot leave the hive during 5 or 6 months of overwintering (cold season). Honey bee don’t defecate in this period. This important factor limits the survival of honey bees in cold environment.

  1. what months? R22: We accept your offer and this information was added in sentence. Please see page 4, line 19. Also, this text presents below: “The overwintering of honey bees lasts for 7 months (from October to April) [100]”

  1. A table can be introduced with columns such as districts, average temperature, Max, min temperatures, overwintering. R23: We thank the reviewer for their comments and have to omit this table in this manuscript for the reason of the massive amount of information about climate and environment in Russian Federation. This information may be the topic next review about environmental conditions for beekeeping in Russian Federation. Because natural resources are the basic background for the lifecycle of honey bees and planning in production beekeeping.

  1. Drone brood stays more time in the colony than worker brood. R24: We thank the reviewer for their comments and changed the sentence.

  1. Reference of honey contamination is more recent than the reference of mites becoming resistant. These 2 statements could be reorganized or the word "recently" replaced by "moreover". R25: We thank the reviewer for their comments and changed the sentence.

  1. It is not clear that how authors use the terminology of this bee as mellifera L. Far Eastern honey bee (Russian bred). The authors must consult literature for writing of name in correct way such as A. mellifera (from Far eastern Russia). The authors must decide and introduce a uniform way to name the Russian bred bees right in the beginning of the description of manuscript. Later, the same common name (with scientific name) should be used throughout the manuscript. For example, A. mellifera (Russian bred from Far-Eastern) or A. mellifera (Russian bred) from Far Eastern Russia or A. mellifera (Far-Eastern Russian bred). The author may adopt their own suitable name. This will omit the confusion being created in the manuscript regarding the name of this honey bee "subspecies" or group. R26: We thank the reviewer for their comments and corrected this basic term. We accepted: inbred bees Apis mellifera Far Eastern breed (Russian-bred) honey bee, initial: Apis mellifera Far East honey bee. This names Apis mellifera Far Eastern honey bee was used in Russian articles. First explanation was done in abstract. Please see page 1, lines 30 - 33. Also, this text presents below: “We illustrate the practical experience of scientists, beekeepers, and breeders in breeding A. mellifera Far East honey bees with Varroa and tracheal mite resistance, which were the initial reasons for breeding the A. mellifera Far Eastern breed by Russian breeders, Russian honey bee in America, the hybrid honey bee in Canada by American breeders, and in China by Chinese beekeepers”.

  1. Thus"! or "However"? the 2 sentences need revision. R27: We thank the reviewer for their comments and we rewrote these sentences. Please see page 12, lines 43 - 47. Also, this text presents below: “Subsequent studies assessed the performance of inbred honey bees. The hygienic behaviour was increased in inbred Russian honey bees (American-bred) in comparison with the domestic North American honey bees [35]. Furthermore, the characteristics of Russian honey bees (American-bred) was performed, marking the differences between inbred Russian and Italian honey bees [137]”.

  1. Indicators R28: We thank the reviewer for their comments and we added in sentence.

  1. need revision. The new breed of honey bees consists of 6 % from total honey bee colonies in Russian Federation R29: We thank the reviewer for their comments and we confirmed this number by unpublished data yet of personal communication Brandorf A.Z.

  1. 30 kg per colony per day? R30: We thank the reviewer for their comments and we applied the citation “Far-Eastern bees are most effective in using the late-summer honey collection from various species of lime, bringing up to 30 kg of nectar per day. Honey productivity is 50–100 kg per bee colony”. Berezin A.S., Borodachev A.V., Borodachev V.A., Mitrofanov D.V., Savushkina L.N. The Loss of Taxonomic Biodiversity of Honey Bees Apis mellifera and Main Breeds in Russia. Phylogenetics of Bees 2019, 144–177, doi:1201/b22405-7.

  1. need revision and clarification. 5.5 thousand bees in each colony? R31: We thank the reviewer for their comments and we changed this sentence. Important part was omitted here: 5 thousand in each colonies of honey bee? Please see page 12, lines 43 - 47. Also, this text presents below: “The number of honey bee colonies of A. mellifera Far Eastern breed (Russian-bred) honey bees is about 5500”.

  1. Volatile? R32: We thank the reviewer for their comments and we changed the word “volatile” to “changeable”.

  1. ...... bees from four breeds and eight bee types originated in the Russian Federation. "Be specific!" Make sure that this is the right way of writing this "group" of bees. I wish the authors discussed why this group of bees wasn't considered a "subspecies" and has a full scientific name. R33: We thank the reviewer for their comments and we clarified this information. Please see page 9, lines 21-26. Also, this text presents below: “The Central Russian ( m. mellifera L.), Carpathian (A. m. carpatica Avet.), Grey Mountain Caucasian (A. m. caucasia Gorb.), A. m. Far Eastern breed (Russian-bred), Bashkirsky breeds, and breed types of A. mellifera L. were included to the State register of Russian bees approved for breeding (http://reestr.gossort.com/reestr/animal/710). The recommendation to breed theirs breed types was made by scientists [15, 81, 82]”.

  1. This needs clarification. Not understood («the morphometric traits were defined to be queens and drones») R33: We thank the reviewer for their comments and we clarified this information. Please see page 12, line 26. Also, this text presents below: “In addition, their morphometric traits were measured and analyzed to characterize the queens and drones [12, 15]”.

  1. Apis mellifera L. in Russian Federation. This portion should be shifted in the beginning as point number 3. This portion is about the introduction of bees and their breeds in Russia. R35: We thank the reviewer for their comments and we shifted this information as chapter 3. Please see page 9.

  1. Add reference R36: We thank the reviewer for their comments and we added references (Borodachev et al., 2019; Berezin et al, 2020).

  1. It is mentioned elsewhere in the ms that Am F Eastern bees is resistant to v mites R37: We thank the reviewer for their comments and we need clarified these sentences. Moreover, the investigations of mellifera Far Eastern bees conducted by scientists of the Research Institute of Beekeeping have not revealed their advantages to Varroa resistance compared to other breeds in Russia (Berezin et al., 2019)” Explanation: Here marked, that other subspecies and types of honey bees in Russia also resistant to varroa. This characteristic possesses no only A. mellifera Far Eastern breed (Russian-bred) honey bee in Russia. This study carried out only scientists of Research Institute of Beekeeping like Krivtsov N. I., Borodachev A.V., Savushkina L.N., Berezin A. S. These articles are in Russian as usual.

  1. Not italicized R38: We thank the reviewer for their comments and we changed.

  1. “light cup of honey cells" needs clarification. R39: We thank the reviewer for their comments and we have to explain this peculiarity. The honey bees different subspecies ( A. m. mellifera, A. m. carpatica, A. m. caucasia) seal the honey cells by the different colour wax cover. This trait was used Russian beekeepers as one of several subspecies’ trait.

  1. Any example of a conference or reference? R40: We thank the reviewer for their comments and we added conference trends to more discussed presentations. Please see page12, lines 29-44. Also, this text presents below: The reasons for honey bee losses and methods of assessment and protection were discussed widely at the International Scientific and Practical Conference on “Modern Problems of Beekeeping and Apitherapy” (Russia, Rybnoye, 2020), and the International conference on “Modern aspects of apitherapy” (Ukraine, Kiev, 2020). The environmental situation has deteriorated around the world during recent years. It was noted that the crossbreeding of honey bees, chemical treatments of plants, insects, and factory emissions have played a major role in this deterioration. The accumulation of metal ions in honey bee bodies has influenced the health of honey bee populations in the Ryazan region [85]. E. K. Eskov, M. D. Eskova (2021) [43] analyzed honey bee losses in polluted areas as a result of the accumulation of pollutants in the honey bee body. Moreover, the performance of honey bees was assessed in terms of hygienic behaviour, which was linked with morphometry characteristics (cubital index, tarsal index, length of proboscis, and the width of tergite 3 and wing veins), honey productivity, and Varroa infestations [11]. In addition, the effectiveness of biotechnological methods has been reported for the prevention of Varroa mites in honey bee populations, including heating and the use of mite traps with the naturally attractive properties of medicinal plants [20].

  1. Any map to show the existence of these places according to the districts as shown in figure 1 and figure 2 R41: We thank the reviewer for their comments and we added the location of breeding apiaries in Figure 2. Please see page 5. Also, this Figure 2 presents below:

  1. List some references!Researchers have developed methods such as biochemistry, behaviour, productivity, morphometric, and genetic to characterize honey bee colonies.” R42: We thank the reviewer for their comments and we added references in sentence. Please see page 5, lines 24-26. Also, this text presents below: “Researchers have developed methods such as assessments of physiology and biochemistry [40, 44, 55, 66, 87, 117], behavior [101, 120], productivity [2, 42, 57, 133], morphometry [18, 19, 24, 126, 147], and genetics [64, 84, 111, 130, 132] to characterize honey bee colonies”.
  2. What are the two subspecies? R43: We thank the reviewer for their comments and we completed the sentence. Please see page 5, lines 29-32. Also, this text presents below: “However, Kirpik et al. (2010) [75] distinguished between m. caucasia and A. m. remipes with the use of a gel electrophoresis method of assessing total protein content. Russian scientists used data on enzyme activity to define the ability of honey bees to survive overwintering [43, 75]”.

  1. Reference R44: We thank the reviewer for their comments and we added references in sentences. Please see page 6, lines 7-9. Also, this text presents below: “Many behavioral traits (swarming, aggressiveness, and others) are used to characterize colonies of honey bees [68] and this is conveniently applied by beekeepers”.

  1. Figure caption needs reformatting. A and B appear as titles. R45: We thank the reviewer for their comments and we changed the title of Figure 3 and removed capture. Please see page 6. 
  2. Was, Were, For, Were R 46: We thank the reviewer for their comments and we changed these mistakes in sentences.

  1. Reference R47: We thank the reviewer for their comments and we rewrote this text. Please see page 7, lines 31-37. Also, this text presents below: “These morphometric traits were chosen for several reasons based on the long-term study of bees [11]. The length of the proboscis was the first trait measured to differentiate between honey bees [3]. The long proboscis was usually used to define the ability of honey bees to collect nectar from deep flowers. Later, the traits from different parts of the body were used to characterize honey bees: the tarsal index, the cubital index, the length of tergite 3, and the length and width of the forewing [5]. Furthermore, these traits have been used to differentiate the subspecies m. mellifera, A. m. carnica, and A. m. caucasia [12, 135]”.

  1. It is unclear that these cluster dendrograms are extracted from previous studies of the author of the present review did the analysis. This should be clarified. If it is based on previous data, references should be added at the end of the figure caption. R48: We thank the reviewer for their comments and we rewrote this text. Please see page 7, lines 3-5. Also, this text presents below: “Two morphometric traits (the length of the proboscis and length of tergite 3) and two indices (cubital and tarsal) of worker bees are presented in Table 1 and used below by us (Figure 4)”.

  1. Were located, Replace by "several", For. R49: We thank the reviewer for their comments and we changed these mistakes in sentences.

  1. Any references? R50: We thank the reviewer for their comments and we rewrote this text. Please see page 7, lines 31-33. Also, this text presents below: These morphometric traits influence the nectar collection ability [77, 134], disease resistance [97], and productivity [96, 98] of honey bees.

  1. Similarly R51: We thank the reviewer for their comments and we removed these mistakes in sentences.

Thank you for accurate explanation of our discrepancies. This manuscript was revised according to your comments. Also, we used the English correction by professional language editing services.

We would like to express our heartfelt gratitude again for the insightful and encouraging comments from the reviewers.

Yours sincerely,

Dr Choi Y. S.

Dr Frunze O.

Reviewer 3 Report

The work of Frunze et al. is interesting and I think that the experience of Russian scientists on breeding and evaluation of honey bees in order to improve the supply of honey bees in the world can be of interest for the scientific public all over the world.

However, the English grammar and the writing style are difficult to follow and they turn their work too hard to read. The authors must devise different ways to show their ideas and results. The sections 3 and 4, between pages 4 to 9, have a lot of information and it is very challenging to read.

The size of the font must be uniformed all over the manuscript.

That is why, and on behalf of the Journal, I suggest accepting after major revisions and all the observations/questions answered.

Author Response

Dear Editor-in-Chief and Reviewer 3,

Thank you for the time and effort spent reviewing our manuscript and suggesting some important points to consider.

We attached the manuscript "Round 1...". The first text (1-21 pages) has no track changes, this text corrected by English editor and me according to Instructions for authors. The second text (22-52 pages) after (below) first text have track changes, but not corrected English editor. Sorry for this complication.

Our replied marked by red colour after symbol (reply) and question number (1-2). For convenience, we copied this text also in response.

Please, find below the reviewer comments (black colour) and our responses (red colour).

Reviewers comments:

  1. However, the English grammar and the writing style are difficult to follow and they turn their work too hard to read. The authors must devise different ways to show their ideas and results. Sections 3 and 4, between pages 4 to 9, have a lot of information and it is very challenging to read. R1 We accepted your suggestions and suggestions of other reviewers, and we did major revisions of our manuscript.
    1. We were revision and rewrote the Title, Simple Summary, Abstract, Introduction, Chapters.
    2. We combined environment information Chapter 2 (first edition) to Chapter 2 Beekeeping Practices in the Russian Federation (last edition).
    3. We moved and renamed Chapter 5 Tendency of Beekeeping in Russian Federation to Chapter 2 Beekeeping Practices in the Russian Federation.
    4. We moved Chapter 4 to Chapter 3, Chapter 3 to Chapter 4.
    5. We corrected the conclusions.
    6. Also, English grammar was checked by professional language editing services recommended by Insects Journal.
  2. The size of the font must be uninformed all over the manuscript R2 We thank the reviewer for their comment and we uniformed our manuscript, including the Figures, Tables, References.

Thank you for your accurate explanation of our discrepancies. This manuscript was revised according to your comments and comments on other reviewers. Also, we used the English correction by professional language editing services.

We would like to express our heartfelt gratitude again for the insightful and encouraging comments from the reviewers.

Yours sincerely,

Dr Choi Y. S.

Dr Frunze O.

Reviewer 4 Report

This review combined the main scientific and productive results regarding the breeding of honey bee stocks in The Russian Federation. I found quite interesting and of scientific value to summarize the knowledge of species, productive and molecular characteristics and the conservation strategies as well. However, as this is presented at the end the reader might get loss during the manuscript reading. I suggest to highlight the history and currently status of the russian honey bee stocks and then include the information about the crossbreeding with american and canadian bees. I also recommend to check for spelling and english language, so the ms woud be more easy to follow. I am not a native speaker as well so I deeply undestand that sometimes writting the ms in a foregein language is quite tricky.  I made some suggestions and included some comments in the PDF file. 

Author Response

Dear Editor-in-Chief and Reviewer 4,

Thank you for the time and effort spent reviewing our manuscript and suggesting some important points to consider.

We attached the manuscript "Round 1...". The first text (1-21 pages) has no track changes, this text corrected by English editor and me according to Instructions for authors. The second text (22-52 pages) after (below) first text have track changes, but not corrected English editor. Sorry for this complication.

Our replied marked by red colour after symbol (reply) and question number (1-29). For convenience, we copied this text also in response.

Please, find below the reviewer comments (black colour) and our responses (red colour).

Reviewers comments:

  1. R1: We thank the reviewer for their comment and we changed the word. Please see page1, line 8.
  2. I think that this sentence should be rewritten. (“conducted an analysis of scientific results of Russian bee research concerning the population study of honey bees over eight districts of the Russian Federation”) R2: We thank the reviewer for their comment and we rewrote the Simple summary. Please see page1. Also, this text presents below:

Simple summary: The loss of honey bees poses a significant problem for the beekeeping industry. Opportunities to recover the stock of honey bees are crucial in areas with drastic losses. In this article, we describe known native and bred Apis mellifera L. stocks of honey bees in the Russian Federation and their characteristics, identified using morphometric and genetic methods. We review the experience of Russian breeders and other breeders in breeding initial A. mellifera Far East honey bees with inherited traits for Varroa resistance. We also describe the bred types of honey bee A. m. mellifera L., A. m. caucasia Gorb., A. m. carpatica Avet. that offer potential in the aim to recover honey bee populations after losses. Our findings indicate that it is necessary to avoid honey bee losses by breeding resistant honey bees. Several bred types of honey bees from the dataset showed high performance in terms of overwintering; resistance to Varroa destructor, Acarapis woodi, and Nosema infections; and little or no swarm conditions. This review shows the potential to increase the selection efforts in the breeding of resistant Apis mellifera L. honey bee populations in the Russian Federation and throughout the world”.

3. This paragraph was not written correctly. (“In this review, methods of recovering honey bee resources have been discussed. The Apis mellifera Far Eastern (Russian bred) bees with a Varroa and Tracheal mite resistance were bred by Russian breeders. Honey bees from Far East were the colony resources for breeding the Russian honey bee (American bred), hybrid honey bee in Canada by American breeders, and in China by Chinese beekeepers. Other new types of honey bees were bred from local colonies ( mellifera mellifera L., A. m. carpatica Avet., A. m. caucasia Gorb.) in breeding apiaries and national parks”). R3: We thank the reviewer for their comment and we rewrote the Abstract. Please see page1. Also, this text presents below: “Here we present an extended literature review and report on personal communications relating to the characterization of the local and bred stock of honey bees in the Russian Federation. New types have been bred from local colonies (A. mellifera L., A. m. carpatica Avet., A. m. caucasia Gorb.). The main selection traits consist of a strong ability for overwintering, disease resistance, and different aptitudes for nectar collection in low and high blooming seasons. These honey bees have been certified by several methods: behavioral, morphometric, and genetic analysis. We illustrate the practical experience of scientists, beekeepers, and breeders in breeding A. mellifera Far East honey bees with Varroa and tracheal mite resistance, which were the initial reasons for breeding the A. mellifera Far Eastern breed by Russian breeders, Russian honey bee in America, the hybrid honey bee in Canada by American breeders, and in China by Chinese beekeepers. The recent achievements of Russian beekeepers may lead to the recovery of beekeeping areas suffering from crossbreeding and losses of honey bee colonies”.

4. resistance ability overwintering. R4: We thank the reviewer for their comment and we corrected this sentence. Please see page1, line 27. Also, this text presents below: “The main selection traits consist of a strong ability for overwintering…”

5. the production? pollination? R5: We thank the reviewer for their comment and we have to explain this sentence. In this case, the contribution is mean the general influence of honey bees on the environment, as the sum of the production of food, pollination.

6. CCD? if so, should be mentioned. Caution because not all losses are due to CCD. R6: We thank the reviewer for their comment and we have to explain our minds. The literature CCD was analyzed as basic in this review. The list of reasons CCD lead to the bee’s mortality is very discussable. We agree with your comment that not all losses are due to CCD. However, the opinion of COLOSS network scientists was supported by Russian beekeepers according to personal communications. We described the problems of honey bee losses, but not discuss the term CCD.

7. Other than parasites and other pathogens R7: We thank the reviewer for their comment and we added the pest of honey bee Megaselia scalaris as an example. Please see page 2, lines 12-21. Also, this text presents below: “Furthermore, the explanation for the decline of honey bees seems to be increasingly attributable not listed above in COLOSS research to Phorid Diptera Phoridae can cause the weakening, reduction, or the disappearance of honey bees, with damage caused by Apocephalus borealis in the United States [31] and by Megaselia ssp. in Europe [112], Africa [27, 93], and Asia [39]. The first warning was reported in Russian Federation in 2014, when the parasite Megaselia scalaris (Loew) was introduced with plants [104]. However, Megaselia ssp. (not Megaselia scalaris) were present in the native environment in the Russian Federation, but their life cycle was restricted by low temperatures [26]. The mass loss of honey bees caused by infestations of Megaselia scalaris has not yet been detected in Russian Federation [110, 118, 144]”.

8. not clear R8: We thank the reviewer for their comment and we rewrote this text. Please see page 3, lines 3-20. Also, this text presents below:

“With the exception of pure local colonies, bee breeders breed new (inbred) types from foreign or local honey bees using selection methods and comparing their traits with initial or domestic colonies [86]. Inbred A. mellifera Far Eastern breed (Russian-bred) honey bees were bred from A. mellifera Far East honey bees [123]. Furthermore, the latter subspecies of honey bees were also the initial colony used for the breeding of Russian honey bees (American-bred) in the USA, hybrids of Russian and Ontario honey bees in Canada [138], and Varroa-resistant A. mellifera Far Eastern honey bees in China (Chinese-bred) [146]. As a consequence, inbred honey bees in the USA, Canada, and China all originated from introduced A. mellifera Far East honey bees. Inbred honey bees have special traits in comparison with domestic colonies. De Gusman et al. (2001) [35] evaluated the effectiveness of A. woodi control treatments in the breeding of Russian honey bees (in inbred honey bees from A. mellifera Far East honey bee). They found that the strong resistance of inbred Russian honey bees (American-bred bees) to tracheal mites was in itself sufficient to minimize tracheal mite damage. Thus, the resistance to Varroa and tracheal mites displayed by the initial honey bees (A. mellifera Far East) were transmitted to Russian honey bees (American-bred) during breeding [114]. Thus, some advantages are displayed by local and inbred honey bees in comparison with foreign colonies”.

9. For, With, Higher, There is an indication. R9: We accept your remark and we changed the sentences.

10. Are these previous results? should be mentioned indicating the reference. If not, results should not be writing here. I am not sure if this is scientific aim R10: We thank the reviewer for their comment and we rewrote this text. Please see page 3, lines 21-26. Also, this text presents below: “This review focuses specifically on known local and inbred types of genetic resources found in honey bees in the Russian Federation which have been identified through the use of various methods (behaviour, genetics, morphometry). The second focus of this paper introduces the results of breeding the same initial mellifera Far East honey bees in different environments: in the Russian Federation, in the USA, in Canada, and in China”.

 11. This is not clear enough (“These harsh climates and other mild climates in south districts of Russian Federation formed the special populations of mellifera L. honey bees.”) R11: We thank the reviewer for their comment and we rewrote this text. Please see page 4, lines 21-25. Also, this text presents below: “The mild climates in the southern part of Russia promote honey bee development and are not described in this review. These climates formed the environments for populations of A. mellifera L. which were detected to possess specific traits and have been used by breeders to breed new types of honey bees in the Russian Federation”.

12. not all climates, and all sub-species. Colonies from temperate colonies need treatment R12: We thank the reviewer for their comment and we rewrote this text. Please see page 11, lines 20-22. Also, this text presents below: “It is not entirely accurate that there have been no bee losses due to Varroa in South America, although Africanized bees, which are present in many areas, are more resistant to the parasite [88]”.

13. morphometric indicator and energy metabolism indicators, as well as DNA markers R13: We thank the reviewer for their comment and we corrected this text. Please see page 13, lines 17-20. Also, this text presents below: “Researchers from Northeast China analyzed the biological characteristics of the introduced and inbred mellifera L. honey bees (Chinese bred), including morphometric indicators and energy metabolism indicators, as well as DNA markers [146]”.

14. maybe this subtitle should go first as this is the original stock, right? R14: We thank the reviewer for their comment and we changed the location of the chapter from 3.2 (second position) to 4.1 (first position). But original stock for all Russian, American, Chinese-bred honey bees was m. Far East honey bee. Please see page 11, line 45. 4.1. Apis mellifera Far Eastern breed (Russian-bred) honey bees.

15. I think that if this is already published, most of it in the same paper, maybe just include the reference but not the table. R15: We thank the reviewer for their comment and we have to apply the table which combines the several morphometric data from the literature. We have to accent the attention on valuable experience to differ the bred types of honey bees by 2 morphometric traits and two indexes.

16. Not clear: “have not revealed their advantages”. R16: We thank the reviewer for their comments and we need to clarify these sentences. Here marked, that other subspecies and types of honey bees in Russia also resistant to Varroa. This characteristic possesses no only mellifera Far Eastern breed (Russian-bred) honey bee in Russia. This study carried out only scientists of Research Institute of Beekeeping like Krivtsov N. I., Borodachev A.V., Savushkina L.N., Berezin A. S. This is an important characteristic of honey bee stock in Russia“.

17. what? This sentence is incomplete R17: We thank the reviewer for their comment and we corrected this text. Please see page 9, lines 35-39. Also, this text presents below: “ m. mellifera Linnaeus 1785 is commonly known as the dark European bee and referred to as the dark bee or the Central Russian bee [59]. This honey bee is selected in breeding apiaries in seven regions of the western part of Russia and is delivered to private apiaries. Therefore, these bees constitute about 60% of the total number of honey bee colonies in the country [84]”.18. stock? R18: We thank the reviewer for their comment and we replied yes. This breed type honey bee was studied and later, there was gain the value as stock.

19. Prikamskaya, Bashkirsky - It not mentioned in the previous paragraph. R19: We thank the reviewer for their comment and we added in the previous paragraph. Please see page 9, lines 48-49; page 10, line 1. Also, this text presents below: “On the basis of m. mellifera honey bees, a new breed (Bashkirsky) and breed types (Prioksky, Orlovskay, Tatarsky, Burzyanskaya bortevaya, and Prikamskaya) of honey bees have been bred and certified.”

20. Created R20: We thank the reviewer for their comment and we changed the word “created” to “selected”.

21. also, selected by a breeding program? R21: We thank the reviewer for their comment and we confirm that breeding program m. caucasia honey bees present in the south part of Russia.

22. I got loss here, as I thought that breeding programs were in order to improve and maintain these traits R22: We thank the reviewer for their comment and we rewrote this text to clarify our minding. Please see page 4, lines 45-49. Also, this text presents below: “The results of the random natural crossbreeding of honey bees led to the loss of purebred honey bee colonies, a decreased ability to survive long overwintering periods, and low resistance to diseases. Preventive actions against the uncontrolled crossbreeding of honey bees in breeding apiaries have not yet been developed at the state level”.

23. provide a favourable environment R23: We thank the reviewer for their comment and we rewrote this sentence. Please see page 4, lines 51-53. Also, this text presents below: “Scientists, in collaboration with employees of national parks, aim to provide the best conditions for the conservation of mellifera L bees. Example of these locations include the Tongariro National Park in New Zealand [99]…”

24. These sentences should be synthesized in one (Swarming leads to the loss of honey bee colonies. It is one of the complex and labor-intensive processes in the apiary (Gaga, Esaulov, 2016). R24: We thank the reviewer for their comment and we rewrote this text to clarify our minding. Please see page 6, lines 4-5. Also, this text presents below: “Swarming is a complex and labour-intensive process occurring in apiaries which leads to the loss of honey bee colonies [47]”.

25. this is incomplete (Many behavioural traits are only apparent at the colony level and therefore characterize the traits of honey bees for convenient breeding by beekeepers) R25: We thank the reviewer for their comment and we rewrote this text to clarify our minding. Please see page 6, lines 7-9. Also, this text presents below: “Many behavioural traits (swarming, aggressiveness, and others) are used to characterize colonies of honey bees [68] and this is conveniently applied by beekeepers”.

26. Probably if you have less colonies in an environment that is offering resources, less colonies may gather more nectar and therefore produce more honey. R26: We thank the reviewer for their comment and we added in sentence. Please see page 6, lines 11-14. Also, this text presents below: “Therefore, when there are fewer colonies in an environment that offers resources for bees, a smaller number of colonies may gather relatively more nectar and therefore produce more honey (Figure 3B). Hence, the performance of honey bees in the Far East was excellent”.

27. this should be included in the figure 3 caption. R27: We thank the reviewer for their comments and we changed the title of Figure 3 and removed capture. Please see page 6. Also, this caption presents below: Figure 3. Honey productivity (A) and Number of bee colonies (B) in eight federal districts in the Russian Federation.

28. Unify with previous paragraph R28: We thank the reviewer for their comment and we rewrote this paragraph. Please see page 6, lines 19-42: page 7, lines 1-2. Also, this text presents below: “The morphometric method was developed by a Russian scientist, Alpatov (1929) [3] Morphometry is a quantitative phenotypic method that analyzes the size of morphological traits at the individual level in honey bees [38]. Ruttner (1988) [116] evaluated bees using thirty-three morphometrics traits, although Russian scientists preferred to measure no more than eight morphometric traits in honey bees: the body color, the length of the proboscis, the width of tergite 3, the length of veins "a" and "b" of the right forewing to calculate the cubital index, the width and length of the metatarsus to calculate the tarsal index, and the discoidal shift of the right forewing [5, 62, 126]. The morphometric traits of local and new breed types of mellifera L. honey bees were analyzed for worker bees, queens, and drones. However, the number of traits to be measured has not been strictly defined by scientists. Morphometric methods have nevertheless been used to distinguish between subspecies and hybrid forms of A. mellifera L. [121, 131, 135, 147]. These morphometric traits were chosen for several reasons based on the long-term study of bees [11]. The length of the proboscis was the first trait measured in order to differentiate between honey bees [3]. The long proboscis was usually used to define the ability of honey bees to collect nectar from deep flowers. Later, the traits from different parts of the body were used to characterize honey bees: the tarsal index, the cubital index, the length of tergite 3, and the length and width of the forewing [5]. Furthermore, these traits have been used to differentiate the subspecies A. m. mellifera, A. m. carnica, and A. m. caucasia [12, 135]. Although the measurement of morphometric traits has been used to separate the subspecies of honey bees, it was not capable of distinguishing between breeds of honey bees [83]. For this reason, the variability of traits was compared. To accomplish this, the absolute value of the coefficient of variation and the standard deviation was calculated [12, 45, 78, 97, 126]. It was also reported that the length and width of the forewing and body characteristics were associated with honey production [42, 98, 134, 145]”.

29. Incomplete R29: We thank the reviewer for their comment and we rewrote this text to clarified our analysis. Please see page 7, lines 19-29. Also, this text presents below: “The m. Far Eastern breed (Russian-bred) honey bees and the original subspecies—the A. m. caucasia and A. m. mellifera bees—were located in different areas on the PCA plot (Figure 4A, C, E). Thus, based on morphometric traits, A. m. Far Eastern breed (Russian-bred) workers, queen bees, and drones were separated from the original sub-species by PCA analysis. However, the hierarchical clustering method revealed another feature. The origin of the A. m. Far Eastern breed (Russian-bred) honey bee from the initial honey bee A. m. caucasia can be observed based on the location of the bees on the same branch of the dendrogram (Figure 4D, F). This feature was present in the analysis of the drones and queens, but not in that of the worker bees. It provides morphometric confirmation of the genetic relationship between A. mellifera Far Eastern breed (Russian-bred) honey bee and the A. m. caucasia honey bee”.

Thank you for your accurate explanation of our discrepancies. This manuscript was revised according to your comments. Also, we used the English correction by professional language editing services.

We would like to express our heartfelt gratitude again for the insightful and encouraging comments from the reviewers.

Yours sincerely,

Dr. Choi Y. S.

Dr. Frunze O.

Round 2

Reviewer 4 Report

Thanks for answering my comments and suggestions.